



Atmospheric
Chemistry
and Physics

# Atmospheric new particle formation characteristics in the Arctic as measured at Mount Zeppelin, Svalbard, from 2016 to 2018

**Haebum Lee[1], Kwangyul Lee[1], Chris Rene Lunder[2], Radovan Krejci[3], Wenche Aas[2], Jiyeon Park[4], Ki-Tae Park[4], Bang Yong Lee[4], Young Jun Yoon[4], and Kihong Park[1]**

[1]School of Earth Sciences and Environmental Engineering, Gwangju Institute of Science and Technology,
123 Cheomdangwagiro, Buk-gu, Gwangju 61005, Republic of Korea
[2]Department for Atmospheric and Climate Research, NILU – Norwegian Institute for Air Research, Kjeller, Norway
[3]Department of Environmental Sciences and the Bolin Centre for Climate Research,
Stockholm University, Stockholm, 106 91, Sweden
[4]Korea Polar Research Institute, 26, Songdo Mirae-ro, Yeonsu-Gu, Incheon, Republic of Korea

**Correspondence:** Kihong Park (kpark@gist.ac.kr) and Young-Jun Yoon (yjyoon@kopri.re.kr)

**Abstract.** We conducted continuous measurements of nanoparticles down to 3 nm size in the Arctic at Mount Zeppelin, Ny Ålesund, Svalbard, from October 2016 to December 2018, providing a size distribution of nanoparticles (3–60 nm). A significant number of nanoparticles as small as 3 nm were often observed during new particle formation (NPF), particularly in summer, suggesting that these were likely produced near the site rather than being transported from other regions after growth. The average NPF frequency per year was 23 %, having the highest percentage in August (63 %). The average formation rate ($J$) and growth rate (GR) for 3–7 nm particles were 0.04 cm$^{-3}$ s$^{-1}$ and 2.07 nm h$^{-1}$, respectively. Although NPF frequency in the Arctic was comparable to that in continental areas, the $J$ and GR were much lower. The number of nanoparticles increased more frequently when air mass originated over the south and southwest ocean regions; this pattern overlapped with regions having strong chlorophyll $a$ concentration and dimethyl sulfide (DMS) production capacity (southwest ocean) and was also associated with increased NH$_3$ and H$_2$SO$_4$ concentration, suggesting that marine biogenic sources were responsible for gaseous precursors to NPF. Our results show that previously developed NPF occurrence criteria (low loss rate and high cluster growth rate favor NPF) are also applicable to NPF in the Arctic.

## 1 Introduction

The Arctic climate system is affected by the region's snow-covered land, sea ice, and ocean, making the region vulnerable to global climate change (Jeffries and Richter-Menge, 2012). Greenhouse gases and aerosols are significant factors affecting the regional climate (Quinn et al., 2007; IPCC, 2014). In particular, aerosols in the ambient atmosphere affect the radiation balance by scattering or absorbing incoming solar light (direct effect) (Toon and Pollack, 1980; Satheesh and Moorthy, 2005) and forming clouds by acting as cloud condensation nuclei (CCN) (indirect effect) (Merikanto et al., 2009).

New particle formation (NPF), which significantly enhances the number of particles in the ambient atmosphere, has been observed in various locations and at various times (Kulmala et al., 2004; Wang et al., 2017; Yu et al., 2017). In favorable conditions, newly formed nanoparticles can, through condensation and coagulation, grow to sizes that allow the formation of CCN. NPF is observed regardless of pollution level, from very clean (e.g., background sites) to heavily polluted (e.g., urban sites), suggesting that various pathways are involved depending on the location and time (Kulmala et al., 2004; Wang et al., 2017). Nucleation can occur almost anywhere in diverse environments, but NPF is observed only when freshly nucleated clusters grow to a detectable size (1–3 nm) (McMurry et al., 2005). Previously de-

veloped criteria for NPF occurrence suggest that a low loss (or scavenging) rate and high growth rate (GR) of clusters increase fresh nuclei survival probability and thus favor NPF, while a high loss rate and low cluster GR suppress it (Kuang et al., 2012).

In the Arctic, a specific phenomenon called "Arctic haze" related to long-range transport of polluted air masses typically occurs in the late winter and early spring (Iziomon et al., 2006; O'Neill et al., 2008, Hirdman et al., 2010). The Arctic haze is associated with elevated concentrations of accumulation-mode particles (Radke et al., 1984; Shaw, 1995; Law and Stohl, 2007; Quinn et al., 2007). A high concentration of accumulation-mode particles results in a high condensational sink (CS) for precursor vapors, which could suppress NPF. The NPF in the Arctic was often reported in summer, when the CS was smaller (Wiedensohler et al., 1996; Covert et al., 1996; Sharma et al., 2013; Willis et al., 2016; Croft et al., 2016). In addition, strong biogenic production from marine and coastal environments in the Arctic region (e.g., Alaska, Alert, and Svalbard) was reported to be linked to NPF due to an increased amount of biogenic sulfur compounds such as dimethyl sulfide and its oxidative products (methane sulfonate and biogenic sulfate) (Leaitch et al., 2013; Park et al., 2017). Like in sulfuric-acid-rich regions, organic-based new particles were observed in pristine environments (Quinn et al., 2002; Karl et al., 2013; Leaitch et al., 2013; Heintzenberg et al., 2015). Asmi et al. (2016) reported that NPF was more common in air masses of oceanic origin compared to continental ones in the Arctic (Tiksi station, Russia). Dall'Osto et al. (2018) suggested that NPF at Station Nord in North Greenland was related to seasonal sea ice cycles (i.e., the NPF was associated with air masses coming from open water and melting sea ice regions).

There are several past studies of NPF at the Zeppelin Observatory at Mount Zeppelin in Svalbard, Norway (Tunved et al., 2013; Dall'Osto et al., 2017; Heintzenberg et al., 2017). The location of the station is 474 m above sea level and ∼ 2 km from a small scientific community, with minimal effects from anthropogenic sources. Its unique geographical location is ideal for investigating NPF in the Arctic environment. Tunved et al. (2013) studied seasonal variations in particle size distribution and NPF based on aerosol size distribution data (10–790 nm) from 2000 to 2010. Heintzenberg et al. (2017) developed a new NPF search algorithm using size distribution data (5–630 nm) from 2006 to 2015. Dall'Osto et al. (2017) determined the relationship between NPF and the extent of Arctic sea ice melt using size distribution data (10–500 nm) from 2000 to 2010 and used hourly data to classify the size distributions and NPF types. It was reported that NPF at the Mount Zeppelin site mostly occurs during summer, which was attributed to the low CS and high biological activity in summer (Leaitch et al., 2013; Heintzenberg et al., 2015; Park et al., 2017). NPF occurrence was low during the Arctic haze (with high CS) period (Tunved et al., 2013; Croft et al., 2016). Heintzenberg et al. (2017)

suggested that NPF at the Mount Zeppelin site was related to solar flux and sea surface temperature, affecting marine biological processes and photochemical reactions with less CS. They reported the potential source regions for NPF to be the marginal-ice and open-water areas between northeastern Greenland and eastern Svalbard. Although particle size distribution data from the Mount Zeppelin site are available (Ström et al., 2003; Tunved et al., 2013; Dall'Osto et al., 2017; Heintzenberg et al., 2017), no data regarding the size distribution of nanoparticles smaller than 5 nm are available, though these could provide greater insight into NPF characteristics. Currently, the initial formation and growth of nanoparticles below 10 nm cannot be resolved, and weak NPF events with no substantial particle growth up to 10 nm cannot be detected.

In this study, we measured number size distribution of nanoparticles down to 3 nm for the first time at Zeppelin station, and we obtained continuous size distributions of 3–60 nm particles every 3 min from October 2016 to December 2018. This allowed the size distribution of nanoparticles to be determined with a lower size limit than before, enabling better identification of whether freshly nucleated particles formed on site or were transported from other regions after substantial growth. We were also able to detect NPF events when particle growth was terminated below 10 nm. The particle size distributions were classified into several clusters, and the seasonal (monthly), daily, and diurnal variations in the nanoparticle concentrations were examined. We also applied the NPF criteria to Arctic data to determine whether or not NPF should occur and investigated the characteristics of NPF events related to formation rate, GR, CS, and meteorological parameters. Finally, potential source regions for NPF were explored using air mass backward trajectory and satellite-derived chlorophyll $a$ concentration data. The chlorophyll $a$, which is involved in oxygenic photosynthesis in the ocean, has been considered a proxy for phytoplankton biomass only. Recent studies showed that there was a strong correlation between sea surface chlorophyll $a$ concentration (estimated by MODIS Aqua) and atmospheric DMS levels at Zeppelin station (Park et al., 2013, 2018).

## 2  Methods

The measurement site was located at the Zeppelin Observatory at Mount Zeppelin, Svalbard (78°54′ N, 11°53′ E), which is 474 m above sea level and ∼ 2 km from the small scientific community in Ny-Ålesund, Norway (78°55′ N, 11°56′ E) (Fig. 1). Ny-Ålesund lies within the west Spitsbergen current at the northernmost point of the warm Atlantic influx; this location provides an ideal location for observing climate parameters and investigating the long-range transport route by which contamination is often carried via southerly air masses (Neuber et al., 2011). The dominant wind patterns (east and southeast from the Kongsvegen glacier (40 %) and

northwest from the Kongsfjorden channels (14 %) during the measurement period from October 2016 to December 2018) and elevation suggest that the effects of local sources on the Zeppelin Observatory are small (Beine et al., 2001).

An air inlet with a flow rate of $100 \, \text{L} \, \text{min}^{-1}$ was used to introduce ambient aerosols into the instruments. The flow temperature was maintained above $0°$ to prevent ice and frost formation in the tube. The observatory was kept warm and dry, with an indoor temperature and relative humidity (RH) of $\sim 20°$ and $< 30 \%$, respectively (Tunved et al., 2013; Heintzenberg et al., 2017). A nano-SMPS (scanning mobility particle sizer) [CE1] consisting of a nano-differential mobility analyzer (nano-DMA) (model 3085, TSI, USA) and an ultra-fine condensation particle counter (model 3776, TSI, USA) was used to measure the size distribution of nanoparticles (3–60 nm) every 3 min; the aerosol flow rate was $1.5 \, \text{L} \, \text{min}^{-1}$ and the sheath flow rate was $15 \, \text{L} \, \text{min}^{-1}$. The size distribution data were processed using the method described by Kulmala et al. (2012).

Daily ionic species ($Na^+$, $Mg^{2+}$, $K^+$, $NH_4^+$, $NO_3^-$, $SO_4^{2-}$, and $Cl^-$) in particulate matter and gas data ($NH_3$ and $SO_2$) at Zeppelin Observatory, along with meteorological parameters (temperature, RH, wind, and pressure), were obtained from the Norwegian national monitoring program (Aas et al., 2019) via the EBAS database (http://ebas.nilu.no/, last access: 2 November 2020). Daily ionic species and gas data are daily measurements collected with a three-stage filter pack sampler (NILU prototype) with no pre-impactor. The size cutoff of the inlet section is approximately $10 \, \mu\text{m}$. Field blanks were prepared the same as the other samples. It should be noted that for the nitrogen compounds the separation of gas and aerosol might be biased due to the volatile nature of $NH_4NO_3$. The detection limits were $0.05 \, \mu\text{g} \, \text{N} \, \text{m}^{-3}$ and $0.01 \, \mu\text{g} \, \text{S} \, \text{m}^{-3}$ for $NH_3$ and $SO_2$, respectively; $0.01 \, \mu\text{g} \, \text{m}^{-3}$ for $Na^+$, $Mg^{2+}$, $K^+$, and $Cl^-$; $0.01 \, \mu\text{g} \, \text{N} \, \text{m}^{-3}$ for $NO_3^-$; $0.05 \, \mu\text{g} \, \text{N} \, \text{m}^{-3}$ for $NH_4^+$; and $0.01 \, \mu\text{g} \, \text{S} \, \text{m}^{-3}$ for $SO_4^{2-}$. The data quality management and system are accredited in accordance to NS-EN ISO/IEC 1702 standards. The detailed information of sampling method and analysis can be found elsewhere (EMEP, 2014; Aas et al., 2019). Solar radiation (SRAD) at the AWIPEV (the Alfred Wegener Institute Helmholtz Centre for Polar and Marine Research and the French Polar Institute Paul Emile Victor) observatory in Ny-Ålesund was obtained from the Baseline Surface Radiation Network (BSRN) (Maturilli, 2019). Hourly data for number size distributions of particles from 5–810 and 10–790 nm, measured with differential mobility particle sizers (DMPSs), were obtained from Stockholm University and the Norwegian Institute for Air Research (NILU), respectively. The data from the DMPS and filter pack measurements are reported to several international monitoring programs (EMEP, European Monitoring and Evaluation Programme; ACTRIS, Aerosols, Clouds and Trace gases Research InfraStructure Network; and GAW-WDCA, Global Atmosphere Watch-the

World Data Centre for Aerosols), and they are openly available from the database infrastructure EBAS. In addition, the hourly black carbon (BC) data at Zeppelin were used to examine the effect of primary combustion sources on the NPF.

Satellite-derived chlorophyll $a$ concentration data in the Svalbard region (70–85° N, 25° W–50° E) were obtained from the level-3 product of the Aqua Moderate Resolution Imaging Spectroradiometer (MODIS) at a 4 km resolution. Air mass backward trajectories arriving at the Zeppelin Observatory were calculated for up to 5 d using the National Oceanic and Atmospheric Administration (NOAA) Hybrid Single Particle Lagrangian Integrated Trajectory (HYSPLIT) model based on Global Data Assimilation System (GDAS) 1° data. A potential source contribution function (PSCF) method (Pekney et al., 2006; Wang et al., 2009; Fleming et al., 2012) was also used to relate the air mass to NPF occurrence by analyzing the residence time of the air mass relative to the concentration of nanoparticles at the receptor site (Wang et al., 2009). In addition, the $k$-means clustering method, an unsupervised data classification and partitioning approach, was used to classify potential air mass origin along with the size distributions (Beddows et al., 2009; Dall'Osto et al., 2017).

The particle GR was calculated as the change rates of representative particle diameters ($d_1$ and $d_2$) with the highest concentrations at particular times ($t_1$ and $t_2$) (Hussein et al., 2005; Kulmala et al., 2012). The CS, which determines how rapidly condensable vapor molecules will condense on the existing aerosols (Kulmala et al., 2012), was calculated from the size distribution data (3–810 nm) with an assumed $H_2SO_4$ diffusion coefficient of $0.117 \, \text{cm}^{-2} \, \text{s}^{-1}$ (Gong et al., 2010; Cai et al., 2017). The number concentration in the size range $d_i$ to $d_j$ ($N_{i-j}$) was derived from the measured size distribution data. Considering the particle loss and production processes allowed the following balance equation for $N_{i-j}$ to be derived:

$$\frac{dN_{i-j}}{dt} = J_{i-j} - F_{\text{coag}} - F_{\text{growth}}, \tag{1}$$

where $J_{i-j}$ is the particle formation rate in the size range of $d_i$ to $d_j$, $F_{\text{coag}}$ is the particle loss rate related to coagulation scavenging in the size range of $d_i$ to $d_j$, and $F_{\text{growh}}$ is the condensational GR of the nucleation-mode particles. Based on methods suggested by Kulmala et al. (2012), the particle formation rate ($J_{i-j}$) was calculated as

$$J_{i-j} = \frac{dN_{i-j}}{dt} + \frac{N_{i-j}}{d_j - d_i} \cdot \text{GR} + N_{i-j} \text{CoagS}_{i-j}, \tag{2}$$

where $\text{CoagS}_{i-j}$ represents the mean of the coagulation sink (CoagS) in the size range of $d_i$ to $d_j$.

The dimensionless criterion ($L_\Gamma$), which can be used to predict the occurrence of NPF events (McMurry et al., 2005; Cai et al., 2017), was calculated as

$$L_\Gamma = \frac{\overline{c}_1 A_{\text{Fuchs}}}{4 \beta_{11} N_1 \Gamma}, \tag{3}$$

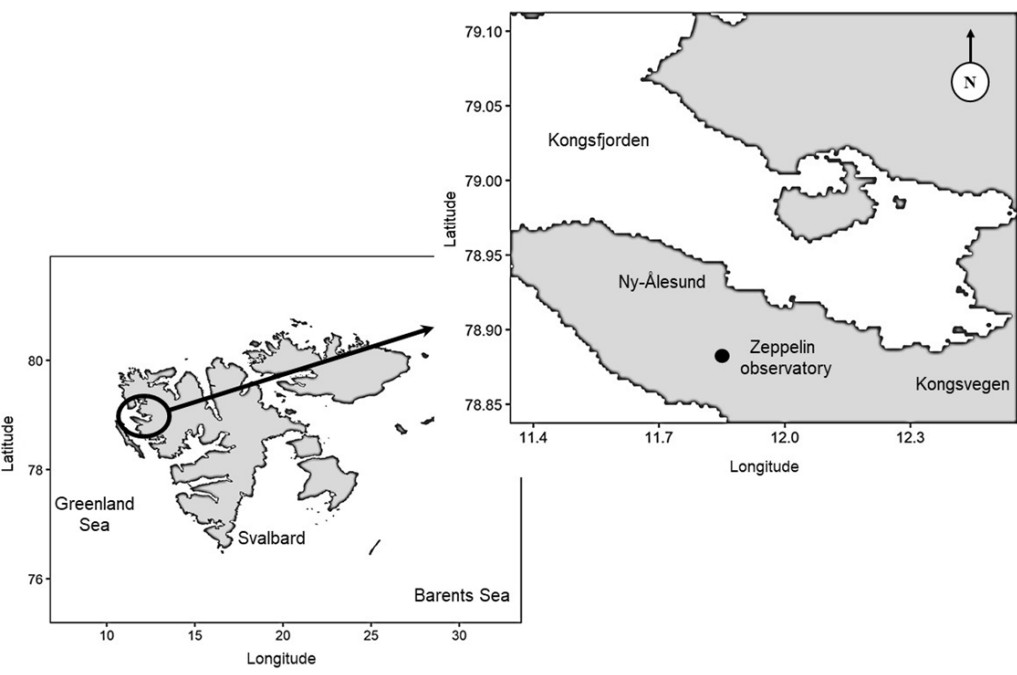

**Figure 1.** Measurement site (Zeppelin Observatory) in the Svalbard Archipelago, Ny-Ålesund, Norway.

where $\bar{c}_1$ is the mean thermal velocity of vapor ($H_2SO_4$), $A_{Fuchs}$ is the Fuchs surface area (a coagulation scavenging parameter), $\beta_{11}$ is the free-molecule collision frequency function for monomer collisions, $N_1$ is the $H_2SO_4$ molecular concentration during the nucleation event, and $\Gamma$ is the growth enhancement factor obtained by dividing the measured GR by the growth determined based on the condensation of only $H_2SO_4$. The $H_2SO_4$ molecular concentration was predicted from the measured daily $SO_2$, hourly CS, hourly solar radiation, and hourly meteorological data (RH and temperature) using the method proposed by Mikkonen et al. (2011). The empirical proxy model of $H_2SO_4$ is given by

$$[H_2SO_4] = a \cdot k \cdot [SO_2]^b \cdot SRAD^c \cdot (CS \cdot RH)^d, \qquad (4)$$

where $[SO_2]$ is the $SO_2$ molecular concentration (molecules cm$^{-3}$), SRAD is the solar radiation (W m$^{-2}$), CS is the condensation sink (s$^{-1}$), RH is the relative humidity (%), and $k$ is the reaction rate constant depending on ambient temperature (see detailed definition for $k$ in Eq. 3 of Mikkonen et al., 2011) with coefficients of $a = 8.21 \times 10^{-3}$, $b = 0.62$, $c = 1$, and $d = -0.13$. The $H_2SO_4$ concentration at Zeppelin was $5.98 \times 10^4$–$3.19 \times 10^6$ molecules cm$^{-3}$ during the summer in 2008 (Giamarelou et al., 2016), which is in a similar range to ours ($2.69 \times 10^4$–$7.68 \times 10^6$ molecules cm$^{-3}$).

## 3   Results and discussion

The data coverage for the size distribution data collected by nano-SMPS was about 89 % during the 27-month sampling period (October 2016 to December 2018). The monthly variations in the number concentrations of the 3–25 nm nanoparticles ($N_{3–25}$) and 25–60 nm nanoparticles ($N_{25–60}$) (averaged from hourly data) are shown in Fig. 2. We compared our nano-SMPS data with DMPS data at the same station as shown in Fig. S1 in the Supplement, suggesting that they were in a good agreement. Both $N_{3–25}$ and $N_{25–60}$ were highest in summer and lowest in winter, indicating that NPF occurred frequently in summer. The higher SRAD and lower CS (calculated from the 3–810 nm size distribution data) in summer also favored nanoparticle production. The highest monthly SRAD (199 W m$^{-2}$) was observed in June. Due to the higher latitude of the site, the SRAD was lower than values reported at other continental sites (449 W m$^{-2}$ during NPF in Lanzhou, China, Gao et al., 2011; TS1 442–445 W m$^{-2}$ during NPF in Pallas, Finland, Asmi et al., 2011; and TS2 700 W m$^{-2}$ during NPF in Atlanta, USA, Woo et al., 2010). The wind speed in summer was lower than in other seasons, as expected from local climatology (Maturilli et al., 2013). In addition, marine biogenic sources, which provide gaseous precursors (e.g., DMS, $H_2SO_4$, and $NH_3$) for nanoparticle formation, were known to be abundant in summer. It was observed that the percentage of air masses passing over high-chlorophyll-$a$ (MODIS data) regions and $H_2SO_4$ and $NH_3$ concentrations measured at the site increased in summer (Fig. S2 and Table S1 in the Supplement). For ex-

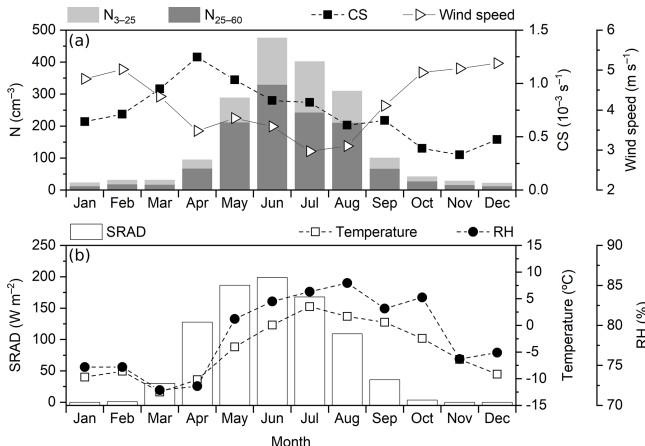

**Figure 2.** Monthly variations in $N_{3-25}$, $N_{25-60}$, CS, and wind speed **(a)**, temperature, RH, and SRAD **(b)** during the measurement period.

ample, chlorophyll *a* concentration (a proxy for marine phytoplankton biomass; Siegel et al., 2013) in the Arctic Ocean surrounding the observation site during the measurement period began to increase in April and reached a maximum in May to June (Fig. S2 in the Supplement). During the Arctic haze period, the number of accumulation-mode particles (> 100 nm) increased considerably. A significant CS increase occurred in March (Fig. 2). The high number of accumulation-mode particles in spring and the high number of nucleation-mode particles in summer are consistent with previous findings (Tunved et al., 2013; Dall'Osto et al., 2017; Heinzenberg et al., 2017).

The size distributions of the 3–60 nm particles during the measurement period (hourly data) were classified into several major groups using the *k*-means clustering method. Four distinct clusters were found (Fig. 3a), with mode diameters of around 10 nm (cluster 1), 20 nm (cluster 2), 30 nm (cluster 3), and 50 nm (cluster 4). Cluster 1 included newly formed particles with high population. Cluster 4 had the lowest ultrafine-particle concentration, representing the background condition. The frequencies of each cluster by month are shown in Fig. 3b. The annual average percentages of each cluster were 7 % (cluster 1), 15 % (cluster 2), 23 % (cluster 3), and 55 % (cluster 4). The frequencies of clusters 1 and 2 increased significantly, and cluster 2 often appeared after cluster 1 in the late spring and summer months (May, June, July, and August), suggesting that strong particle growth (i.e., increases in mode diameter) after NPF occurred during those months.

We identified two distinct types of NPF (Fig. 4). In type 1, $N_{3-25}$ increased significantly with subsequent particle growth (the freshly formed particles experienced gradual growth), a typical banana-shaped nucleation event, which is regularly observed at many locations worldwide. In type 2, $N_{3-25}$ increased significantly without clear subsequent particle growth (almost no increase in the mode diameter with

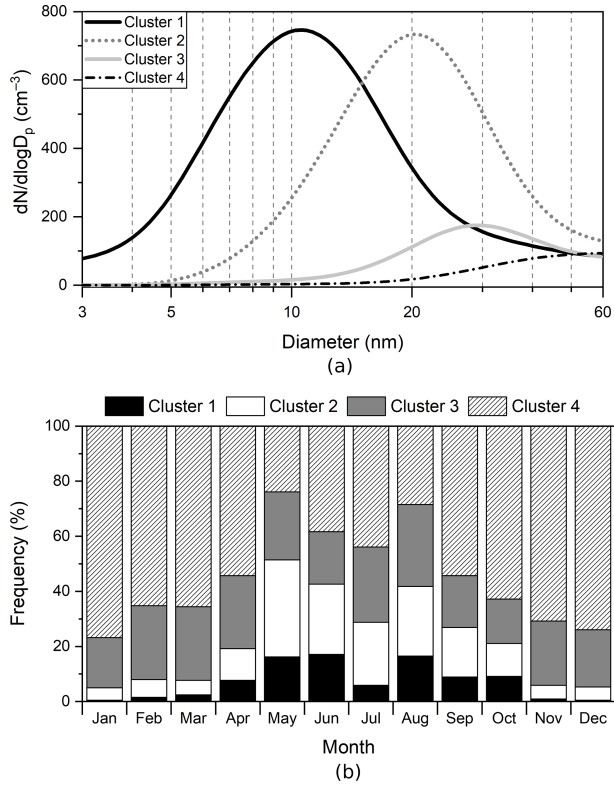

**Figure 3.** Major particle clusters by **(a)** size distribution and **(b)** monthly frequency of clusters during the measurement period.

time, or not clear for growth); this type of event lasted more than 2 h. Therefore, the GR could be calculated only for type 1. The cases not matching either of these were classified as "undefined" NPF, for which $N_{3-25}$ increased for a short period of time (less than 2 h). This NPF classification approach was similar to methods employed previously (Dal Maso et al., 2005; Kulmala et al., 2012; Nguyen et al., 2016). The mean occurrence percentage of NPF days (all types) per year from the measurement period was 23 %. Dall'Osto et al. (2017) found that the average of yearly NPF occurrence from 2000 to 2010 was 18 %, lower than our value, and that this increased over time as the coverage of sea ice melt increased. Based on the Heintzenberg et al. (2017) study, the mean occurrence percentage of NPF days per year from 2006 to 2015 was estimated to be around 20 %. In addition, DMS originating from marine sources can be a key precursor contributing to NPF in the remote marine atmosphere (Leaitch et al., 2013; Park et al., 2017; Jang et al., 2019). In the Arctic region, the DMS concentration increased by 33 % per decade from 1998 to 2016 (Galí et al., 2019), potentially leading to the increase in the annual NPF occurrence in this area.

It was shown that the concentration of fine particles could be affected by local combustion sources such as local port and cruise ships (Eckhardt et al., 2013). The effects of anthropogenic sources (e.g., downtown, local port, and cruise

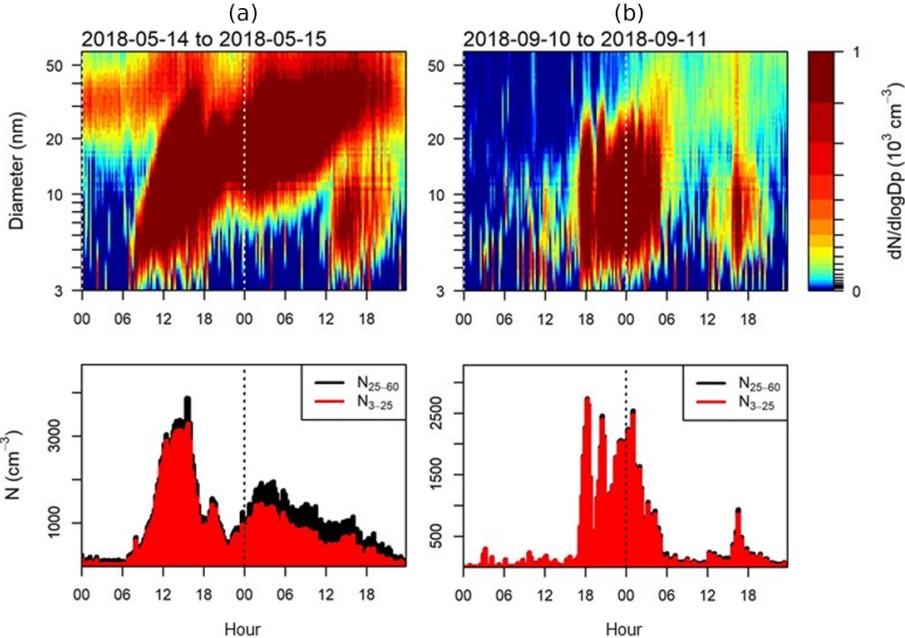

**Figure 4.** Examples of distinct NPF types identified in this study. In type 1 **(a)**, $N_{3-25}$ increases significantly with continuous particle growth, while in type 2 **(b)** it increases significantly without significant particle growth. The $x$ axis is the local time (hour).

ship) on the NPF were examined by using local wind and air mass trajectory data to find whether air mass or wind passed over the Ny-Ålesund downtown and local port during NPF events. Also, the concentration of black carbon (BC) at Zeppelin, typically emitted from primary combustion sources, was used to examine the effect of primary combustion sources on the NPF. We found that the air mass and wind passed over the downtown including the local port during only two NPF events out of all NPF events (170 events). During these two NPF events, the BC concentration increased little. Thus, we believe the effect of anthropogenic sources on the NPF should be small. Also, in our NPF data analysis we filtered out two NPF events with increased BC concentration and wind direction coming from the Ny-Ålesund downtown or port.

The highest percentage of NPF occurrence for all types was observed in August (63 %) and June (61 %), followed by May (47 %) and July (42 %) as shown in Fig. 5. NPF was observed only occasionally in winter during the Arctic night from November to February, consistent with previous observations (Ström et al., 2009; Heintzenberg et al., 2017). Although NPF occurrence could be expected to be lowest in April due to the highest CS (Fig. 2), that was not the case. Our results showed that NPF occurrence increased significantly in April, was maintained at a high level from May to August, and then decreased in September and October. The average values of CS during NPF event and non-event days were $0.57 \times 10^{-3}$ and $0.69 \times 10^{-3}$ s$^{-1}$, respectively. The higher biological and photochemical activity, lower transport of pollutants from midlatitudes, and increased wet scav-

enging of particles (low CS) in summer likely favored NPF (Ström et al., 2009). In addition, the melting of sea ice in summer can increase the availability of marine biogenic sources, promoting NPF (Quinn et al., 2008; Tovar-Sánchez et al., 2010; Dall'Osto et al., 2018). Overall, NPF occurrence is mainly affected by the availability of solar radiation (photochemistry) and gaseous precursors in addition to the survival probability of clusters or particles (Kulmala et al., 2017). In addition, it was suggested that fragmentation of primary marine polymer gels, which are derived from phytoplankton along the marginal ice zone, could be a source for atmospheric nanoparticles (NPF events below 10 nm) in the high Arctic boundary layer (Heintzenberg et al., 2017; Karl et al., 2019; Mashayekhy Rad et al., 2019).

A so-called "weak NPF" event, in which initial formation and growth were completed to < 10 nm without further growth, was observed. The weak NPF events documented here could not be detected in previous studies where the minimum detectable size was ∼ 10 nm. The fraction of weak NPF occurrences (out of all NPF occurrences each month) was highest in April (58 %) and October (50 %), compared to values in May (20 %), June (14 %), July (8 %), August (15 %), and September (25 %). In April, this was likely caused by the combination of strong solar radiation (i.e., strong photochemistry for production of condensing vapors responsible for particle growth) and high CS; in contrast, October's combination of the low solar radiation (i.e., weak photochemistry) and low CS led to a similar result.

NPF lasted for several hours with similar start times (Fig. 5). NPF duration was around 6–7 h on average and was

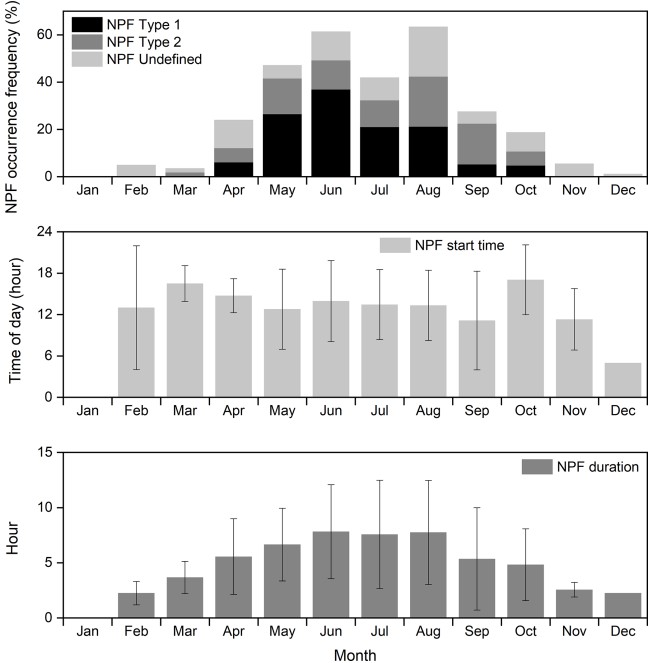

**Figure 5.** Monthly variations in NPF occurrence, start time (local time), and duration; the error bar represents standard deviation.

longest in summer. Typically, NPF started between 13:00 and 14:00 (local time), suggesting that photochemical activity with strong solar radiation played an important role in NPF initiation. The variations in start time from month to month (March to November) were smaller than the monthly variations in NPF occurrence or duration. The nighttime NPF also occurred in late fall to winter (20 % out of total NPF events). The exact mechanism for this NPF was unclear. Nanoparticles formed at earlier times (daytime) in other places may be transported to the site during nighttime (Vehkamäki et al., 2004; Park et al., 2020).

Figure 6 shows the MODIS monthly chlorophyll *a* concentrations around Svalbard, which increased from April and decreased after August. The chlorophyll *a* concentration was intense in the ocean regions southwest and southeast of Svalbard. A recent study revealed that the DMS production capacity of the Greenland Sea (to the southwest) was 3 times greater than that of the Barents Sea (to the southeast) (Park et al., 2018); this is further discussed in the context of air mass trajectory data in a later section. Full monthly values of average chlorophyll *a* concentration over the area (70–85° N, 25° W–50° E) and "air mass exposure to chlorophyll *a*" ($E_{chl}$), which explains the DMS mixing ratio of the air mass arriving at Zeppelin (Park et al., 2018), are summarized in Fig. S2 in the Supplement. The $E_{chl}$ provides the measure of potential DMS production capacity of the ocean air mass passed over (Park et al., 2018). It was found that "air mass exposure to chlorophyll *a*" ($E_{chl}$) was correlated well ($r = 0.69$ and $p$ value $< 0.05$; not shown) with the NPF oc-

currence frequency, compared to the average chlorophyll *a* concentration over the area (70–85° N, 25° W–50° E).

To determine the characteristics of particle growth, we calculated the GR in the 3–7, 7–25, and 3–25 nm size ranges (i.e., $GR_{3-7}$, $GR_{7-25}$, and $GR_{3-25}$) for NPF events (Fig. 7). The average $GR_{3-25}$ for all months was 2.66 nm h$^{-1}$, comparable to previously reported GR data (0.2–4.1 nm h$^{-1}$) in the Arctic region (Kerminen et al., 2018). The highest monthly average $GR_{3-25}$ was observed in July (3.03 nm h$^{-1}$) and the maximum individual value (6.54 nm h$^{-1}$) occurred in June. The averages of $GR_{3-7}$ and $GR_{7-25}$ were 2.07 and 2.85 nm h$^{-1}$, respectively. However, the GR was much lower than the values observed in typical urban areas (Table 1), suggesting a lower availability of condensing vapors contributing to particle growth in the Arctic atmosphere. The formation rates of particles in the same size range as calculated GR were also derived. The averages of $J_{3-7}$, $J_{7-25}$, and $J_{3-25}$ during NPF events were 0.04, 0.09 and 0.12 cm$^{-3}$ s$^{-1}$, respectively. The highest monthly average and maximum for $J_{3-7}$ were both found in June but for $J_{7-25}$ and $J_{3-25}$ were found in July. The formation rates (relative standard deviation (RSD) = 39 %–44 %) varied by month more significantly than for GR (RSD = 27 %–33 %). The formation rates in this study were much lower than those reported in continental areas (Stanier et al., 2004; Hamed et al., 2007; Wu et al., 2007; Manninen et al., 2010; Xiao et al., 2015; L. Shen et al., 2016; Cai et al., 2017). A good linear relationship was found between $J_{3-7}$ and $N_{3-7}$ ($r = 0.97$ and $p$ value $< 0.001$) as shown in Fig. S3 in the Supplement, indicating that 3–7 nm particles were produced by gas-to-particle conversion rather than direct emissions in the particle phase (i.e., not primary) (Kalivitis et al., 2019). No significant correlation was found between $J_{3-7}$ and $GR_{3-7}$, suggesting that the vapors participating in the early stage of NPF could be at least partly different from the vapors contributing to subsequent particle growth (Nieminen et al., 2014). However, detailed chemical data for nanoparticles during formation and growth should be obtained to achieve complete understanding of the participating chemical species. Our data indicate that, although NPF occurrence frequency in the Arctic was comparable to continental areas, the $J$ and GR were much lower. Time series of daily GR and $J$ in different modes ($GR_{3-7}$ and $J_{3-7}$ and $GR_{7-25}$ and $J_{7-25}$), weekly $N_{3-7}$ and $N_{7-25}$, and weekly NH$_3$ and H$_2$SO$_4$ are shown in Fig. S4 in the Supplement.

The existence of significant amounts of nanoparticles as small as 3 nm during NPF events at the study site suggests that NPF occurred there, rather than the particles being transported from other regions after growth. In other words, if NPF occurred at other locations far from the study site, the nanoparticles would have grown during transport to the site and few 3 nm particles would have been detected there. The lifetime of the 3 nm particles in this study (growth to particles larger than 7 nm) was estimated to be 2–3 h on average. It was reported that nanoparticles (< 5 nm) in the troposphere

**Table 1.** Summary of NPF frequency, $J$, and GR at various sampling sites, including the present study. NA means not available.

| Site name and characteristics | | Period | NPF frequency | GR (nm h$^{-1}$) | | $J$ (cm$^{-3}$ s$^{-1}$) | | Reference |
|---|---|---|---|---|---|---|---|---|
| Zeppelin, Norway | Arctic | October 2016 to December 2018 | 23 % | GR$_{3-7}$<br>GR$_{7-25}$<br>GR$_{3-25}$ | 0.29–5.17<br>0.45–6.94<br>0.48–6.54 | $J_{3-7}$<br>$J_{7-25}$<br>$J_{3-25}$ | 0.001–0.54<br>0.003–0.50<br>0.007–0.61 | This study |
| Finokalia, Greece | Marine background | June 2008 to June 2018 | 27 % | GR$_{9-25}$ | 5.4 ± 3.9 | $J_{9-25}$ | 0.9 ± 1.2 | Kalivitis et al. (2019) |
| Beijing, China | Urban | March 2004 to February 2005 | 40 % | GR$_{3-25}$ | 0.1–11.2 | $J_{3-25}$ | 3.3–81.4 | Wu et al. (2007) |
| Pittsburgh, USA | Urban | July 2001 to June 2002 | 30 % | NA | NA | NA | NA | Stanier et al. (2004) |
| San Pietro Capofiume, Italy | Sub-urban | March 2002 to March 2005 | 36 % | GR$_{3-20}$ | 2.9–22.9 | $J_{3-20}$ | 0.2–36.9 | Hamed et al. (2007) |
| 12 European sites (EUCAARI project)* | Rural and background | 2008 to 2009 | 21 %–57 % | GR$_{7-20}$ | 3.6–6.8 | $J_{2-3}$ | 0.7–32.4 | Manninen et al. (2010) |
| Hyytiälä, Finland | Rural | 1996 to 2003 | > 24 % | GR$_{3-25}$ | 0.9–5.3 | $J_{3-25}$ | 0.2–1.1 | Dal Maso et al. (2005) |
| Shangdianzi station, China | Rural | March 2008 to December 2013 | 36 % | GR$_{3-25}$ | 0.7–13.4 | $J_{3-25}$ | 0.5–39.3 | X. Shen et al. (2016) |
| Pyramid, Nepal | Himalayas | March 2006 to August 2007 | > 35 % | GR$_{10-20}$ | 1.8 ± 0.7 | $J_{10-20}$ | 0.05–0.2 | Venzac et al. (2008) |
| Dome C | Antarctica | December 2007 to November 2009 | 5 %–54 % | GR$_{10-25}$ | 0.5–4.6 | $J_{10-25}$ | 0.022–0.11 | Järvinen et al. (2013) |
| Neumayer | Antarctica | January 2012 to March 2012, February 2014 to April 2014 | NA | GR$_{3-25}$ | 0.4–1.9 | $J_{3-25}$ | 0.02–0.1 | Weller et al. (2015) |
| King Sejong | Antarctica | March 2009 to December 2016 | 6 % | GR$_{10-25}$ | 0.02–3.09 | $J_{2.5-10}$ | 0.16–9.88 | Kim et al. (2019) |
| Nord, Greenland | Arctic | July 2010 to February 2013 | 17 %–38 % | NA | NA | NA | NA | Nguyen et al. (2016) |

\* Pallas and Hyytiälä (Finland), Vavihill (Sweden), Mace Head (Ireland), Cabauw (Netherlands), Melpitz and Hohenpeissenberg (Germany), K-Puszta (Hungary), Jungfraujoch (Switzerland), Puy de Dome (France), San Pietro Capofiume (Italy), and Finokalia (Greece).

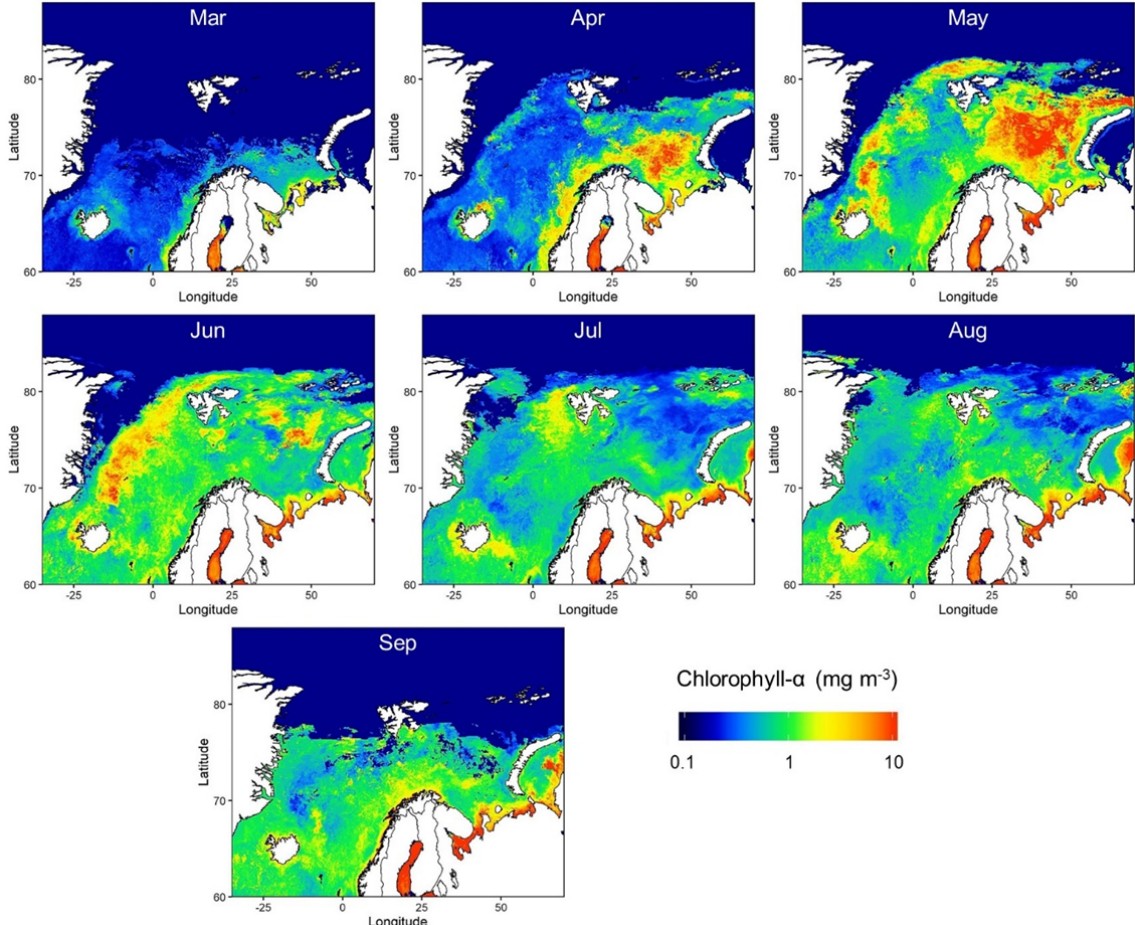

**Figure 6.** MODIS-derived monthly chlorophyll *a* concentration during the measurement period at 4 km resolution.

could survive for several hours or less (Anastasio and Martin, 2001).

Five air mass clusters were found (Fig. 8a), representing the contributions of different air masses in different seasons: clusters 1, 2, 3, 4, and 5 represented southwest (slow), south (slow), southeast (fast), northwest (fast), and northeast (fast) air masses, respectively. The air mass speed (travel distance divided by time) was used to determine whether the air mass was slower or faster compared to the average air mass speed during the measurement period. Cluster 1 dominated in summer, when NPF occurrence was highest; it had the lowest air mass speed, the lowest fraction of land influence (15 %), and the highest fraction of time spent over the sea (50 %) compared to other air mass clusters. Time spent over sea ice was 35 %. The CS values were $0.54 \times 10^{-3}$, $0.74 \times 10^{-3}$, $0.77 \times 10^{-3}$, $0.64 \times 10^{-3}$, and $0.80 \times 10^{-3}$ s$^{-1}$ for cluster 1, cluster 2, cluster 3, cluster 4, and cluster 5, respectively, suggesting that cluster 1 had the lowest CS. Our data suggest that a slowly moving air mass, which spent most of the time over the ocean and sea ice, is the most favorable for NPF.

We further explored the potential source regions of the air masses in relation to NPF using air mass backward trajectory data and the 75th percentile of $N_{3-25}$ (Fig. 8b). Increases in the amount of nanoparticles (i.e., NPF events) occurred more frequently when the air mass passed over the oceanic regions to the southwest and south of Svalbard (overall, 49 % of the air mass during NPF was southwest, i.e., cluster 1). As shown earlier (Fig. 6), the chlorophyll *a* concentration was strong in the southwest and southeast ocean regions, and the DMS production capacity of the southwest ocean was 3 times greater than that of the southeast ocean. The DMS production capacity was defined as the potential amount of DMS produced from the phytoplankton biomass (Park et al., 2018). Several previous studies also support the strong DMS production capacity in the southwest ocean (Degerlund and Eilertsen, 2010; Galí and Simó, 2010). These results suggest that marine biogenic sources from the southwest ocean (Greenland Sea) region play an important role in NPF in the Arctic.

The DMS in the ocean is produced by complicated microbial food-web processes (Stefels et al., 2007). In gen-

https://doi.org/10.5194/acp-20-1-2020

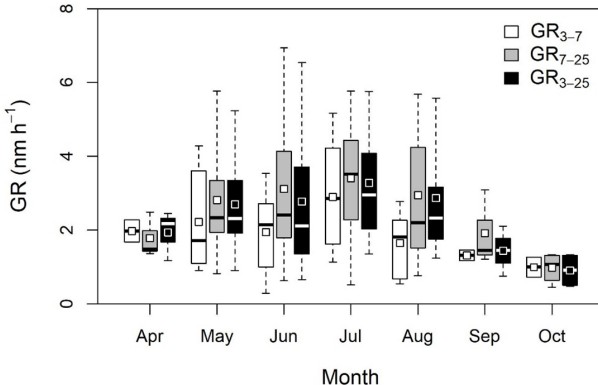

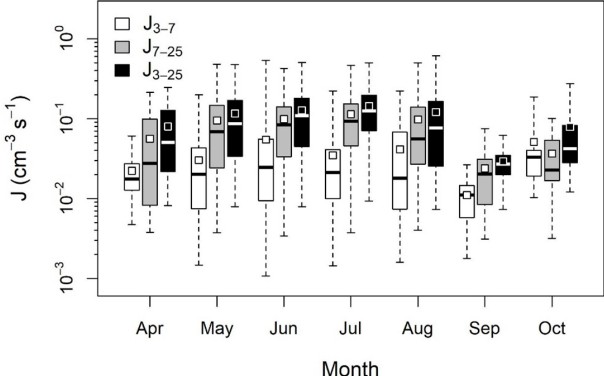

**Figure 7.** Monthly variations in $GR_{3-7}$, $GR_{7-25}$, $GR_{3-25}$, $J_{3-7}$, $J_{7-25}$, and $J_{3-25}$ for NPF in the Arctic. Boxes and whiskers represent the 25th–75th percentiles and minimum–maximum, respectively; squares indicate means and horizontal lines within boxes indicate medians.

eral, sea surface DMS maximum occurs following local phytoplankton biomass maxima, thereby leading to lag periods on the order of several weeks to months (so-called DMS summer paradox) (Galí and Simó, 2015). This phenomenon could be explained by several key processes: a succession in phytoplankton composition, grazing by zooplankton on dimethylsulfoniopropionate-containing (DMSP-containing) CE2 phytoplankton and the bacterial degradation of DMSP into DMS (Polimene et al., 2012). However, a clear temporal correlation between atmospheric (and/or seawater) DMS level and phytoplankton biomass (i.e., chlorophyll $a$ concentration) has been observed for the ocean domains where strong DMS producers (both containing high intra-cellular DMSP content and DMSP cleavage enzyme) such as haptophytes and dinoflagellates predominate (e.g., Arnold et al., 2010; Park et al., 2013, 2018; Uhlig et al., 2019; Zhang et al., 2020). Only a limited number of phytoplankton classes including dinoflagellates and haptophytes possess the enzyme that can convert DMSP into DMS during their growth (Alcolombri et al., 2015). In particular, *Emiliania huxleyi* and *Phaeocystis* sp., which are highly abundant haptophytes in high-latitude oceans, play key roles in con-

trolling global DMS emission because the DMS production capacity of these species is much higher than other globally abundant phytoplankton species (Liss et al., 1994; McParland and Levine, 2019). For example, multi-year measurements of atmospheric DMS mixing ratios at Zeppelin station showed a strong correlation between sea surface chlorophyll $a$ concentration (estimated by MODIS Aqua) and atmospheric DMS levels (Park et al., 2013, 2018). Furthermore, relationships between the atmospheric DMS and phytoplankton biomass were regionally and temporally varied with the relative abundance of strong DMS(P) producers (Park et al., 2018). This is because the oceanic DMS production in the vicinity of the observation site (i.e., Greenland and Barents seas) was largely governed by direct DMS exudation of phytoplankton that have both high cellular DMSP content and the DMSP-cleavage enzyme during the phytoplankton bloom period. A recent study conducted at a remote Antarctic site also revealed that the number concentration of nano-size particles (3–10 nm in diameter) was positively correlated with the chlorophyll $a$ concentration during the period when strong DMS producers predominate (dominance of *Phaeocystis* > 50 %; estimated by PHYSAT algorithm) (Jang et al., 2019).

We then examined the chemical characteristics of particulate matter (PM) and daily concentration of gaseous $NH_3$. The seasonal characteristics of ionic species ($Na^+$, $Mg^{2+}$, $K^+$, $NH_4^+$, $NO_3^-$, $SO_4^{2-}$, and $Cl^-$) in PM during the measurement period (Table S1 in the Supplement) revealed that the contributions of primary sea salt particles ($Na^+$, $Mg^{2+}$, and $Cl^-$) increased in winter with high wind speeds, while the contributions of $NH_4^+$, $NO_3^-$, and $SO_4^{2-}$ (secondary species) increased in spring and summer. The slope of the cation equivalents ($Na^+$, $Mg^{2+}$, $K^+$, and $NH_4^+$) versus the anion equivalents ($NO_3^-$, $SO_4^{2-}$, and $Cl^-$) ($= 0.98$; not shown) suggested that the measured cations were mostly neutralized by the anions (Zhang et al., 2015). These ionic species can exist in large particles and do not necessarily represent the chemical composition of the nanoparticles, but they can provide information about the overall chemical properties of the particles in different seasons. The non-sea-salt sulfate (nss-$SO_4^{2-}$) could have had a secondary origin from the DMS from the sea (Park et al., 2017; Kecorius et al., 2019). The $SO_4^{2-}$ could also come from sea salt particles (primary production of $SO_4^{2-}$) (Karl et al., 2019). Thus, the concentration of nss-$SO_4^{2-}$ was derived from nss-$SO_4^{2-}$ ($\mu g\,m^{-3}$) = total $SO_4^{2-}$ ($\mu g\,m^{-3}$) $- 0.252 \times Na^+$ ($\mu g\,m^{-3}$) by using the measured $SO_4^{2-}$ and $Na^+$ concentrations (Zhan et al., 2017). The nss-$SO_4^{2-}$ ratio (nss-$SO_4^{2-}$ / total $SO_4^{2-}$) was significantly higher on NPF event days than on non-event days ($p$ value $< 0.01$; Fig. 9). The $NH_3$ concentration was higher on NPF event days than on non-event days as shown in Fig. 9 ($p$ value $< 0.001$), similar to results shown in Dall'Osto et al. (2017), although daily $NH_3$ concentration was not significantly correlated with the $N_{3-25}$ as shown in Fig. S5 in the Supplement.

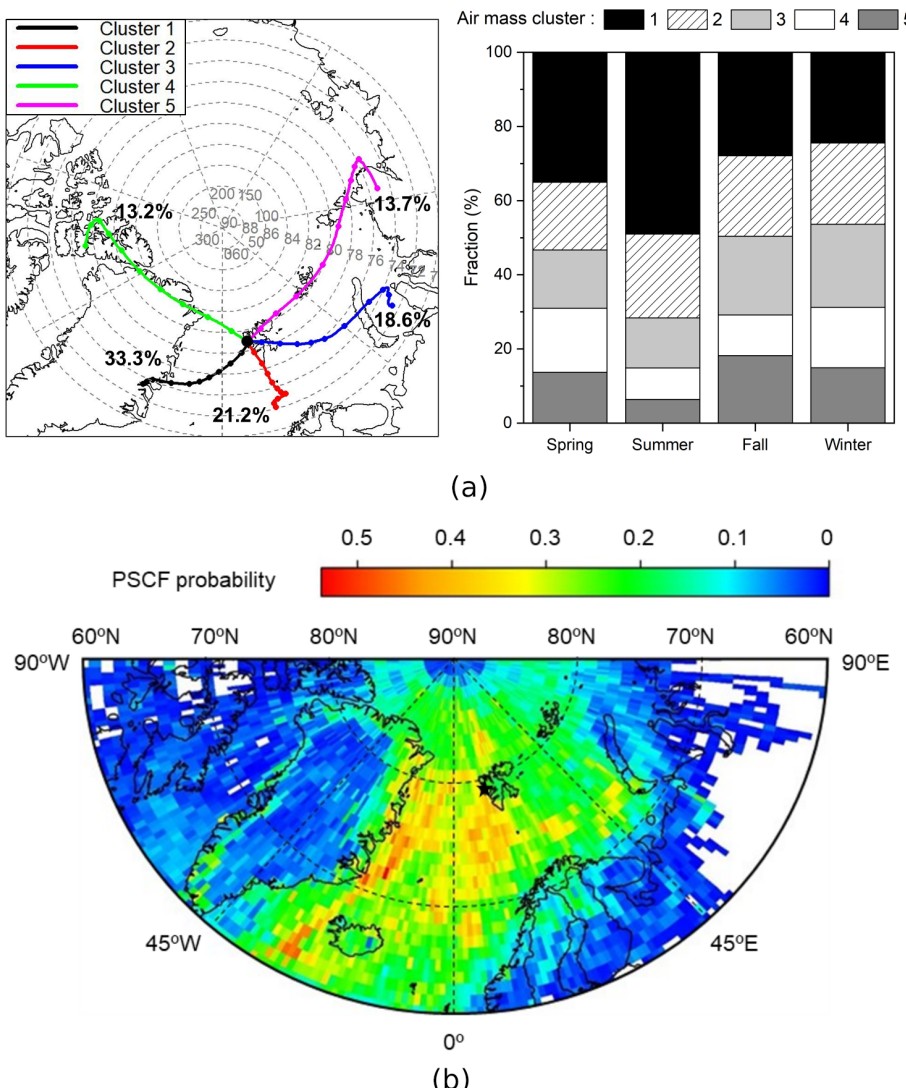

**Figure 8. (a)** Five major clusters for air mass back trajectories during the measurement period and the fraction of each cluster by seasons. **(b)** PSCF back-trajectory analysis for air mass origins affecting NPF at the 75th percentile of $N_{3-25}$.

The NH$_3$ in the Arctic can originate from biological and animal sources (e.g., seabird colonies) (Tovar-Sánchez et al., 2010; Croft et al., 2016; Dall'Osto et al., 2017). The SO$_2$ was not significantly higher on NPF event days than on non-event days (Fig. 9) and not significantly correlated with the $N_{3-25}$ (Fig. S5 in the Supplement). On the other hand, the H$_2$SO$_4$ was found to be higher on the NPF event days (Fig. 9) and was correlated with the $N_{3-25}$ (Fig. S5 in the Supplement), suggesting that the H$_2$SO$_4$ should play an important role in nucleation and growth. Our data were limited to fully explain the nucleation mechanism. Further studies should be required to elucidate the nucleation mechanism by directly measuring chemical composition of nanoparticles and various precursor vapors.

The NPF event probability distribution with daily CS and temperature was included in Fig. S6 in the Supplement. The

NPF event probability was calculated by the ratio of the NPF event days per total days for the given CS and temperature. The NPF event probability increased at moderate temperatures when the CS was low, while when the CS was high, the probability increased at relatively high temperatures as shown in Fig. S6 in the Supplement.

We calculated the NPF criterion ($L_\Gamma$) values for NPF event and non-event days (Fig. 10). The 7 non-event days when GR could be obtained from pre-existing aerosols were selected for the calculation of the $L_\Gamma$ (Kuang et al., 2010). The NPF duration time was determined using the proposed method (Kulmala et al., 2012), with the time range of non-event days set as daytime (06:00–18:00 LT). When NPF occurred, the $L_\Gamma$ ranged from 0.003 to 0.27 with a mean and median of 0.044 and 0.041, respectively; all values were less than 1. The $L_\Gamma$ values of non-event days ranged from 0.34 to 2.59

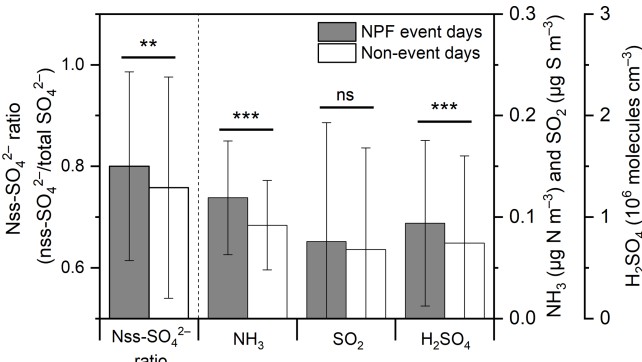

**Figure 9.** Comparison of average nss-$SO_4^{2-}$ ratio (nss-$SO_4^{2-}$ / total $SO_4^{2-}$), $NH_3$, $SO_2$, and $H_2SO_4$ concentrations between NPF events and non-event days: error bar and stars represent the standard deviation and $p$ values of a $t$ test (ns: $> 0.05$, *: $\leq 0.05$, **: $\leq 0.01$, ***: $\leq 0.001$), respectively.

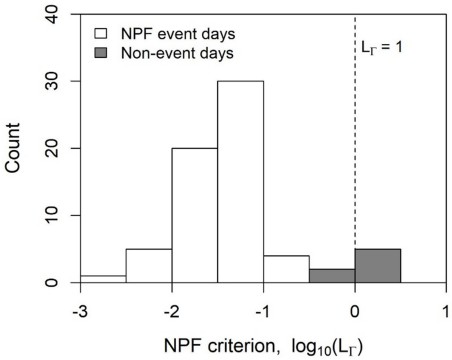

**Figure 10.** Distribution of NPF criterion ($L_\Gamma$) values for NPF event days (white) and non-event days (grey) in the Arctic.

with a mean and a median of 1.49 and 1.61, respectively; 5 d were larger than 1. These observations were consistent with previous studies of NPF events in clean or moderately polluted areas (Tecámac, Atlanta, Boulder, and Hyytiälä), ranging from 0.0075 to 0.66 (Kuang et al., 2010), and in a highly polluted area (Beijing), ranging from 0.22 to 1.75 (Cai et al., 2017). Our data suggest that $L_\Gamma$ can also be useful for determining the occurrence of NPF in the Arctic, but not at 100 % certainty. Uncertainties in $H_2SO_4$ concentration inferred from daily $SO_2$ data (as discussed in the experimental section) and other parameters such as the measured GR and averaging time for $L_\Gamma$ (i.e., NPF duration time) could contribute to unclear separation of NPF event and non-event days (Fig. 10).

## 4 Conclusions

We examined the characteristics of Arctic NPF at the Mount Zeppelin site by conducting continuous measurements of nanoparticles down to 3 nm in size from October 2016 to

December 2018. The size distributions of 3–60 nm particles were classified into distinct clusters with strong seasonal variability and mode diameters of 10 nm (cluster 1), 20 nm (cluster 2), 30 nm (cluster 3), and 50 nm (cluster 4). A significant number of nanoparticles as small as 3 nm often appeared during NPF, particularly in summer, suggesting that there is a good chance that these were produced near the site rather than being transported from other regions after growth. The average NPF occurrence frequency per year was 23 %. $J_{3-7}$ averaged 0.04 cm$^{-3}$ s$^{-1}$, ranging from 0.001 to 0.54 cm$^{-3}$ s$^{-1}$, and GR$_{3-7}$ averaged 2.07 nm h$^{-1}$, ranging from 0.29 to 5.17 nm h$^{-1}$. These data suggest that the NPF occurrence frequency in the Arctic is comparable to that in continental areas, although the $J$ and GR were lower in the Arctic. We next identified five major air mass clusters using backward-trajectory analysis; PSCF results indicated that air masses from the south and southwest ocean regions were related to the elevated concentrations of nanoparticles at the site. This region was consistent with elevated chlorophyll $a$ and DMS production capacity, suggesting that marine biogenic sources should play an important role in Arctic NPF. The concentrations of $NH_3$ and $H_2SO_4$ were higher on NPF event days than on non-event days. Previously developed NPF criteria (a low ratio of loss rate to growth rate of clusters favors NPF) were applicable to Arctic NPF occurrence.

*Data availability.* The nano-SMPS data (3–60 nm) in 2016 to 2018 are available on the Korea Polar Data Center (KPDC) website (https://doi.org/10.22663/KOPRI-KPDC-00001127.2 TS3, https://doi.org/10.22663/KOPRI-KPDC-00001125.3 TS4, https://doi.org/10.22663/KOPRI-KPDC-00001126.4 TS5, Nano-SMPS particle number concentration, Park and Lee, 2020 TS6), and the raw data can be distributed upon request to the corresponding author (kpark@gist.ac.kr). The DMPS (5–810 and 10–790 nm) data are available from the Stockholm University and Norwegian Institute for Air Research (NILU) and also via the EBAS database (http://ebas.nilu.no TS7, last access: 2 November 2020). The meteorological data for solar radiation (SRAD) were provided by the Alfred Wegener Institute (Maturilli, 2019).

*Supplement.* The supplement related to this article is available online at: https://doi.org/10.5194/acp-20-1-2020-supplement.

*Author contributions.* HL and KL applied the statistical methodology and generated results. HL, CRL, RK, WA, and KTP analyzed the results. HL, KL, CRL, WA, RK, JP, and KTP participated in the field measurements and collected the data. KP, YJY, and BYL designed the study. HL and KP prepared the manuscript with contributions from all co-authors.

*Competing interests.* The authors declare that they have no conflict of interest.

*Acknowledgements.* This research was supported by a National Research Foundation of Korea grant from the Korean Government (Ministry of Science and ICT) (NRF-2016M1A5A1901779) (KOPRI-PN20081) (Title: Circum Arctic Permafrost Environment Change Monitoring, Future Prediction and Development Techniques of Useful Biomaterials, CAPEC project), and a National Leading Research Laboratory program (NRF-2019R1A2C3007202). We also would like to thank research engineers Tabea Henning, Ondrej Tesar, and Birgitta Noone from ACES and the staff from the Norwegian Polar Institute (NPI) for their on-site support. NPI is also acknowledged for substantial long-term support in maintaining the measurements at Zeppelin Observatory. We also would like to acknowledge the support by the Samsung Advanced Institute of Technology (SAIT), the long-term support of the Swedish EPA's (Naturvårdsverket) Environmental Monitoring Program (Miljöövervakning), the Knut and Alice Wallenberg Foundation within the ACAS project (Arctic Climate Across Scales, project no. 2016.0024), and FORMAS (project no. 2016-01427). TS8

*Financial support.* This research has been supported by National Research Foundation of Korea grants from the Korean Government (grant nos. NRF-2016M1A5A1901779 and TS9 NPF-2019R1A2C3007202).

*Review statement.* This paper was edited by Veli-Matti Kerminen and reviewed by three anonymous referees.

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

**Remarks from the language copy-editor**

CE1    The definition was left here as it is the first instance of the term within the text.

CE2    Please note the changes.

**Remarks from the typesetter**

TS1    According to our standards, changes like this must first be approved by the editor, as data have already been reviewed, discussed and approved. Please provide a detailed explanation for those changes that can be forwarded to the editor. Please note that this entire process will be available online after publication. Upon approval, we will make the appropriate changes. Thank you for your understanding.

TS2    Please see previous remark regarding editor's approval.

TS3    Please provide a reference list entry including creators, title, and repository.

TS4    Please provide a reference list entry including creators, title, and respository.

TS5    Please provide a reference list entry including creators, title, and respository.

TS6    Please provide corresponding reference.

TS7    Please provide a direct link to the data set, if possible. In any case, please provide a reference list entry including creators and title.

TS8    Please note that all funding information should be part of the financial support section according to our standards. We allow the funding information to be included in both the acknowledgements and the financial support section if you would like to leave the acknowledgements section as it is, or the information can be moved from the acknowledgements to the financial support section. Please let us know how you would like to proceed. Thank you.

TS9    Please check: you corrected this number in the acknowledgements but not here.