# Peer review of "Atmospheric new particle formation characteristics in the Arctic as measured at Mount Zeppelin, Svalbard, from 2016 to 2018"

_Atmospheric Chemistry and Physics, 2020_

## Referee Comment (RC1) · Anonymous Referee #1 · 13 May 2020

The authors present beutiful data on NPF in the Arctic. They showed that the NPF events were correlated with DMS production and NH3 concentration. I agree importance of those chemical components for NPF events in the Arctic region. The paper is well written and it can be accepted after some revisions and discussions. I have a comment on the precursor of the NPF event.

1) Is it possible to add analysis on the correlations between NPF and SO2 gas concentration (and other possible precursors)?

---

## Short Comment (SC1) · 16 Jun 2020

Interesting paper, however, I have some comments regarding the origin of the particles and/or precursors.

The authors claim, that due to the small size occasionally observed the particles have to be produced in the vicinity of the Zeppelin station. That's plausible. However, in that case it would be interesting whether anthropogenic emissions for example from the port may contribute. The frequency of particle events fits well into the frequency of ships in Ny-Alesund. That port and ships affect the site is clearly shown for example in Eckhardt et al, 2013, here also the typical meteorological conditions for such an

anthropogenic contamination at Zeppelin.

The event day May 14, 2018 is one of these days with low and variable winds at the Zeppelin site discussed in the Eckhardt paper (see also the PANGANEA meteorological data for this day). Also HYSPLIT, when calculated with the higher resolution of 0.5 degree instead of the default GDAS 1 degree shows an air mass trajectory sweeping over the Kongsfjord and the port of Ny-Alesund while the 1 degree resolution does not show this Ny-Alesund loop. A more detailed local meteorology would be very helpful in the search for the origin of nucleation mode particle precursors.

Second, the presence of cruise ships is roughly the same like the frequency of particle events during the summer months (May to August). These ships stay in the port normally for the daylight hours and leave the port at 16:00 local time. However they are often in the fjord even for a longer time span. For the port there is the harbor keeping a record. For the whole archipelago the AIS database (www.marinetraffic.com) may be used. Ships are emitting a huge amount of ultrafine and even nucleation mode particles, especially when they are equipped with catalysts for NOx removal in emission control areas, see for example Kivekäs et al, 2014 or Kecorius et al, 2016. A single cruise ship in the vicinity (and up to 50 NM upwind) thus might be a dominating source of either primary nanoparticles or of nanoparticle precursors, especially in the otherwise pristine environment of Spitzbergen where normally sulphur compounds are thought to originate from DMS (open sea) and ammonia from seabird colonies (islands).

It would be good when such anthropogenic contamination could be excluded. Enclosed is the HYSPLIT analysis for May 14, 3 trajectories, two hour intervals

References Eckhardt et al. Atmos. Chem. Phys., 13, 8401–8409, 2013, www.atmos-chem-phys.net/13/8401/2013/, doi:10.5194/acp-13-8401-2013

Kivekäs et al, 2014, Atmos. Chem. Phys. Discuss., 14, 8419–8454, 2014, www.atmos-chem-phys-discuss.net/14/8419/2014/doi:10.5194/acpd-14-8419-2014
Simonas Kecorius et al, Significant increase of aerosol number concentrations in air masses crossing a densely trafficked sea area, Oceanologia, Volume 58, Issue 1, 2016, Pages 1-12, ISSN 0078-3234, https://doi.org/10.1016/j.oceano.2015.08.001.
* * *
[Figure]

[Figure]

05/14/2018 0500 UTC

Mt_Zeppelin_obs

05/14/2018 0700 UTC

05/14/2018 0900 UTC

Image Landsat / Copernicus
Image IBCAO

10.2 km

Goo

**Fig. 1.**

---

## Referee Comment (RC2) · Anonymous Referee #2 · 17 Jun 2020

"Atmospheric new particle formation characteristics in the Arctic as measured at Mount Zeppelin, Svalbard, from 2016 to 2018" by Haebum Lee et al.

General:

To assess the effect of ocean biology on atmospheric new particles formation observed between 2016-2018 at the Mount Zeppelin, Svalbard, this study was set to explore (using MODIS monthly mean satellite Chlorophyll-a concentrations) whether any direct link or correlation exist between ocean biological sources and the observed frequency of occurrence of nanoparticles as small as 3 nm diameter.

It was concluded that nanoparticles increased more frequently when the origin of

air masses reaching the Zeppelin observatory overlapped with regions having strong Chlorophyll-a concentration and dimethyl sulfide (DMS) production capacity, and was also correlated with increased daily NH3 concentrations. Moreover, the authors argue that the primary drivers of the observed new particles formation are the seasonal cycles of ocean biological activity or presence of sea bird colonies. However, it's essential to keep in mind that overlap or correlations do not prove a causal relationship. Unfortunately, the authors do not present any assessment of the reliability and credibility of the derived correlation.

Moreover, the literature survey is not fully convincing; there is some previous observational evidence on new particle formation from the Zeppelin observatory that reflect the seasonal cycle of gel-forming marine microorganisms and their controlling factors, that would seem to have an essential bearing on the results obtained and, thus, appear to merit discussion. Examples are Heintzenberg et al., 2017; Karl et al., 2019 and Mashayekhy Rad et al., 2019.

Furthermore, the satellite derived Chlorophyll-a numbers used in this study cannot be used in order to demonstrate a biological source in terms of biological activity (productivity/bloom) or phytoplankton primary productivity for the following reason; Chlorophyll-a is used a proxy of phytoplankton biomass whereas a phytoplankton bloom in various phases is the combined effect of phytoplankton production and zooplankton grazing, cell lysis and/or bacterial degradation. To be able to make any kind of statement relating the biological activity, phytoplankton biomass- and speciation, zooplankton etc. to DMS water concentrations, data on these parameters are required, covering the period 2016-2018.

The generally, over marine areas, poor correlation observed between parallel measurements of seawater DMS and phytoplankton biomass (Chlorophyll-a) has been explained as a consequence of the species specificity of DMS production followed upon the production of from extra-cellular (dimethyl sulfonium propionate) DMSPp.

A further very important reason for the in general poor correlation between DMS in the water and in algae biomass is that DMS is produced from DMSPd and not from intra-cellular DMSPp. This means that the production and breakdown of DMS in the water column must be looked on as a result of complex physiological and ecological interactions, as demonstrated in Leck et al. (1990). To be able to make any kind of statement linking the biological productivity, phytoplankton biomass and DMS water concentrations, data on the above-mentioned parameters are all required. As no such measurements were performed it is not valid to extrapolate results of high satellite derived Chlorophyll-a concentrations in the surface waters to DMS in the water column which will be subsequent emitted into the atmosphere and there undergo photochemical oxidation to form various intermediate products and, ultimately, sulphuric acid promoted nucleation.

Therefore, the statements causally linking the effect of ocean biology or as referred to in this study "DMS production capacity" on atmospheric new particles formation observed at the Zeppelin observatory are not well founded or supported and should be removed or de-emphasized.

As the conclusions are based on essential inputs of MODIS monthly mean satellite Chlorophyll-a concentrations and the suggested presence of bird colonies, which are both not constrained by measured in-situ data and that it seems that the authors have a somewhat limited conceptional understanding of the detailed processes involved needed for a successful assessment the effect of ocean biology on new particle formation observed at the Zeppelin observatory, I could only recommend the manuscript for publication after major revision according to the given comments and suggestions.

Although many of the above-identified parameters are not given as an integrated part of the present study, the manuscript has an important value as a descriptive data report of the seasonal life cycle of nanometer sized particles down to 3 nm diameter (with a relatively high temporal resolution), a parameter in general sparsely measured in the marine environment and specifically so in the Arctic.

Detailed:

Page 2, line 50: relevant reference to add is Karl et al., 2013. Page 2, line 63: Please omit Covet et al., 1996. Page 3, line 68: Please add Heintzenberg et al., 2017. Page 3, line 72: Please insert number between measured and size. To not confuse the reader be specific on that you discuss size distributions by number, throughout the manuscript. Page 3, line 78: Please remove the superscript (10) on "occur".

Page 3, line 80: Please motivate why you will use satellite-derived Chlorophyll-a concentration data to detect potential source regions for new particle formation. Please also add that Chlorophyll-a is a proxy of phytoplankton biomass only.

Page 3, lines 90-93: Observations are a challenge and specifically so in the pristine remote Arctic marine environment. Specific to the Zeppelin observatory extreme care must be exercised to prevent interference from local pollution (ship traffic) and thus contamination of the air samples. Please add information on the implemented procedure to detect and avoid contamination by local pollution or from long-range transport (southerly air mass origin). Did you have any automatically interruption of the sampling when necessary, due to unfavorable conditions (pollution sensor)?

Page 4, line 97: General to the manuscript. Please replace specie(s) with compounds or constituents as specie(s) belongs to the family of living organisms.

Page 4, line 97: All Ionic – and molecular formulas should be define.

Page 4, lines 97-101: Please give the 50% cut-off equivalent aerodynamic diameters (EADs) of the 3-stage filterpack sampler (type?). Also give details on the analytical methods used for both the particular matter and gas phase compounds collected. How were the blank levels determined? Analytical detection limits obtained for the various ions? Were any Quality checks of the IC-analyses performed? LOD, precision?

Page 4, line 102: Please define AWIPEV.

Page 4, line 106: Please define EMEP, ACTRIS, GAW-WDCA.

Page 5, line 142: Please add details on how the sulfuric acid number concentration was predicted from the measured daily SO2, please also discuss the quality of the data in use.

Page 5, line 154: Be specific, which are the precursor gases in mind? How do you verify their abundance? How will a proxy measure of marine phytoplankton biomass (Chlorophyll-a) influence the availability of atmospheric precursor gases for new particle formation?

Page 6, line 176: Please add the results by Heintzenberg et al., 2017.

Page 6, line 178: Please clarify which results that support the following statement "In addition, DMS originating from marine sources can be a key precursor contributing to NPF in the remote marine atmosphere".

Page 6, lines 181-182; Page 7, lines 203-204 : How do you explain the poor correlation observed between the highest percentage of new particle formation (Fig. 5), and the relatively low MODIS monthly mean satellite-derived phytoplankton biomass (Chlorophyll-a) concentrations in the month of August (Fig. S1 bottom and Fig.6)?

Page 6, line 187: I cite "In addition, the melting of sea ice in summer can increase the availability of marine biogenic sources, promoting NPF". Please specify which the sources you have in mind? Here I find the literature survey unconvincing; there is quite a large amount of previous observational evidence that would seem to have an essential bearing on the results obtained and, thus, appear to merit discussion. In this respect, it seems somewhat surprising that even not mention of or learn from the previous work by Leck and her colleagues over the last three decades on releases of atmospheric sulfur compounds and marine sea-spray aerosols (organic polymer gels/inorganic) over the Arctic pack ice area (incl. the marginal ice zone) in summer.

Page 7, lines 204-207; Page 8 lines 239-242; Page 9, line 281: Figure 6 shows that the Chlorophyll-$\alpha$ concentrations were most pronounced in the ocean areas southwest and

southeast of Svalbard. After the authors explored the potential source regions of the air masses in relation to occurrence of nanoparticles, it was found that increasing numbers of the latter occurred more frequently when the air trajectories passed over the oceanic regions to southwest and south of Svalbard but surprisingly not when passing over the ocean areas south east of Svalbard. This was explained by that the DMS production capacity of the southwest ocean was 3 times greater than that of the southeast ocean. To support the findings the authors used the results derived from a study based on data sets obtained between May and April in 2010, 2014 and 2015 by Park et al. (2018). Please give a detailed explanation to what "DMS production capacity" stands for and what it critically depends on. Please also explain how the findings by Park et al., 2018 covering only two months (April and May) of the biological season (this study April to October) and in different years, could be used to explain the findings in this study.

Page 7, line 208: Please support your statement: "The existence of significant amounts of nanoparticles as small as 3 nm during NPF events at the study site suggests that NPF occurred there, rather than the particles being transported from other regions after growth." What is the expected atmospheric residence time of the nucleated particles?

Page 8, line 252: Please clarify the meaning of that non-sea-salt sulfate could have a secondary origin from oceanic DMS. Which formula was used in the calculations of non-sea-salt sulfate. How do you estimate the contribution from non-biogenic DMS sources? A much more strait forward comparison would be to use particulate methane sulfonate (MSA) concentrations of the total suspended particle samples.

Page 8, line 252: Could you please discuss how realistic your assumption on a DMS derived (sulfuric acid) nucleation mechanism is in respect to the findings by Pirjola et al. (2000), which showed that, under typical conditions in the MBL, homogeneous binary $H_2SO_4$-$H_2O$ nucleation will not occur and ternary $H_2SO_4$-$H_2O$-$NH_3$ nucleation will only be sufficiently effective to produce observable particles for DMS concentrations in the range of 400 ppt(v) or higher and very low aerosol condensation sinks.

Page 18, Figure 2: Please remove the superscript (37) on "SRAD".

References (a selection)

Bigg, E.K., C. Leck and L. Tranvik, 2004. Particulates of the surface microlayer of open water in the central Arctic Ocean in summer, Marin Chemistry, 91, 131-141.

Bigg, E.K., and C. Leck, 2008, The composition of fragments of bubbles bursting at the ocean surface, J. Geophys. Res., 113 (D1), 1209, doi:10.1029/2007JD009078.

Gao, Q., C. Leck, Rauschenberg, and P.A. Matrai, 2012, On the chemical dynamics of extracellular polysaccharides in the high Arctic surface microlayer, Ocean Sci., 8, 401-418.

Hamacher-Barth, E., C. Leck, and K. Jansson, 2016, Size-resolved morphological properties of the high Arctic summer aerosol during ASCOS-2008, Atmos. Chem. Phys., 16, 6577-6593, doi:10.5194/acp-16-6577-2016.

Heintzenberg, J., C. Leck, and Tunved, P., 2015, Potential source regions and processes of the aerosol in the summer Arctic, Atmos. Chem. Phys., 15, 6487-6502, doi:10.5194/acp-15-6487-2015.

Karl, M., C. Leck, F. Mashayekhy Rad, A. Bäcklund, S. Lopez-Aparicio, J. Heintzenberg, 2019, New insights in sources of the sub-micrometre aerosol at Mt. Zeppelin observatory (Spitsbergen) in the year 2015, Tellus B, 71 (1), 1-29.

Kerminen, V-M., and C. Leck, 2001, Sulfur chemistry over the central Arctic Ocean in summer: Gas to particulate transformation, J. Geophys. Res., 106 (D23), 32,087-32,099.

Leck, C., U. Larsson, L.-E. Bågander, S. Johansson, and S. Hajdu, 1990, DMS in the Baltic Sea-Annual variability in relation to biological activity, J. Geophys. Res. 95, 3353-3363.

Leck, C., and C. Persson, 1996a, Seasonal and short-term variability in dimethyl sul-

fide, sulfur dioxide and biogenic sulfur and sea salt aerosol particles in the arctic marine boundary layer, during summer and autumn, Tellus 48B, 272-299.

Leck, C., and C. Persson, 1996b, The central Arctic Ocean as a source of dimethyl sulfide - Seasonal variability in relation to biological activity, Tellus 48B, 156-177.

Leck, C. and K. Bigg, 2007, A modified aerosol–cloud–climate feedback hypothesis. Environmental Chemistry, 4, 400–403, doi:10.1071/EN07061.

Leck, C., and E.K. Bigg, 2005a, Biogenic particles in the surface microlayer and overlaying atmosphere in the central Arctic Ocean during summer, Tellus 57B, 305-316.

Leck, C., and E.K. Bigg, 2005b, Evolution of the marine aerosol – A new perspective. Geophys., Res., Lett., 32, L19803. doi: 10.1029/2005GL023651.

Mashayekhy Rad, F., J. Zurita, P. Gilles, L.A.J. Rutgeerts, U. Nilsson, Leopold L.I., C. Leck, 2019, Measurements of Atmospheric Proteinaceous Aerosol in the Arctic Using a Selective UHPLC/ESI-MS/MS Strategy, Journal of The American Society for Mass Spectrometry, 30(1), 161-173.

Orellana M.V., P.A. Matrai, C. Leck, C. D. Rauschenberg, A. M. Lee, and E. Coz 2011, Marine microgels as a source of cloud condensation nuclei in the high Arctic, PNAS, 108 (33): 13612-13617.

Pirjola, L., O'Dowd, C. D., Brooks, I. M., and Kulmala, M., Can new particle formation occur in the clean marine boundary layer?, J. Geophys. Res., 105(D21), 26,531-26,546, 2000.

---

## Referee Comment (RC3) · Anonymous Referee #3 · 24 Jun 2020

Review of 'Atmospheric new particle formation characteristics in the Arctic as measured at Mount Zeppelin, Svalbard, from 2016 to 2018' by Lee et al.

The manuscript studies the characteristics of NPF at Mount Zeppelin, a location in the Arctic far from direct anthropogenic emissions. The study compromises ∼2 years of comprehensive valuable data suitable for NPF study. While NPF has been studied at the same location, the new data included in this study contains high time resolution of particle number size distributions of particle sizes relevant for new particle formation. The manuscript is well written, the methods used are clearly described and the literature review is thorough. I suggest publication in ACP after addressing the comments

below.

General comments:

1. The exact dates of the measurements need to be reported to identify the reoccurrence of the NPF seasons. The authors mention '89% during the 27 months sampling period', but the exact months need to be mentioned.

2. I agree with Anonymous Referee #1 on the necessity of showing the correlation between concentrations of precursor vapours and particle formation rates and growth rates. How do the concentrations of these vary between event days and nonevent days. Something like your figure 9 would be nice to show also for gas-phase precursors. It could be divided into monthly event days and non-event days.

3. The trends of the precursor vapors during the measurement period (sulfuric acid and ammonia), the number concentrations in different clusters, and different modes (3-7 nm, 7-25 nm) as well as the particle formation and growth rates can be shown as daily or weekly medians, maybe in the supplementary. Similar to Kalivitis et al. (2019) figures 8c and 8d or Mikkonen et al. (2020) figure 2.

4. You calculate J3-7 but GR3-25, although the GR is not constant over the size bin 3-25 nm (Kulmala et al., 2013). Calculating a size segregated GR, i.e. GR3-7 and GR7-25 is recommended especially looking at your figure 4 (upper left), the GR is not constant over these sizes.

5. I don't understand why you chose to present the data in UTC and not Local time. When using UTC, there is no relevance to solar radiation or to other locations. Please show your figures in local time (Figure 4 and Figure 5-middle). You can also show figure 5-middle relative to sunrise. See for example figure 6 in Dada et al. (2018).

6. What about nighttime clustering? your figure middle panel shows that the start time of NPF is around 20 UTC? also unit of time needs to be added to the figure or caption.

7. There seems to be an effect of temperature as well as CS on the probability of NPF.
See figure 13 in Dada et al. (2017). Does the occurrence of Arctic haze inhibit 3 nm clustering and growth? How different is the CS between NPF event days and non event days? If possible, you could examine how CS varies between the airmass clusters.

8. Why GR3-25 while N3-20? Maybe use 3- 25 nm as nucleation mode for consistency with your GR calculations and with previous literature. N3-25 has been referred to as nucleation mode particles in some literature (Vana et al., 2016; Zhou et al., 2020).

9. Comparison of instruments: how does your nano-smps compare to the instruments at the station? See figure 1 in Kangasluoma et al. (2020).

Specific comments:

Line 14: " a higher resolution than ever before", this sentence needs to be changed since previous studies have shown up to 10 s time resolution, unless you mean at the specific location you are measuring. Please change here and else where.

Line 36, anytime, do you mean anytime during the month? or anytime during the day? because very little nighttime NPF that grows to full NPF events are observed in the boundary layer.

Line 38, survivor —> survival

Line 175, 'Dall Maso' –> Dal Maso.

Line 190, survivable –> survival

References:

Dada, L., Paasonen, P., Nieminen, T., Mazon, S. B., Kontkanen, J., Perakyla, O., Lehti-palo, K., Hussein, T., Petaja, T., Kerminen, V. M., Back, J., and Kulmala, M.: Long-term analysis of clear-sky new particle formation events and nonevents in Hyytiala, Atmos Chem Phys, 17, 6227-6241, 10.5194/acp-17-6227-2017, 2017.

Dada, L., Chellapermal, R., Buenrostro Mazon, S., Paasonen, P., Lampilahti, J., Man-

ninen, H. E., Junninen, H., Petäjä, T., Kerminen, V. M., and Kulmala, M.: Refined classification and characterization of atmospheric new-particle formation events using air ions, Atmos. Chem. Phys., 18, 17883-17893, 10.5194/acp-18-17883-2018, 2018.

Kalivitis, N., Kerminen, V. M., Kouvarakis, G., Stavroulas, I., Tzitzikalaki, E., Kalkavouras, P., Daskalakis, N., Myriokefalitakis, S., Bougiatioti, A., Manninen, H. E., Roldin, P., Petäjä, T., Boy, M., Kulmala, M., Kanakidou, M., and Mihalopoulos, N.: Formation and growth of atmospheric nanoparticles in the eastern Mediterranean: results from long-term measurements and process simulations, Atmos. Chem. Phys., 19, 2671-2686, 10.5194/acp-19-2671-2019, 2019.

Kangasluoma, J., Cai, R., Jiang, J., Deng, C., Stolzenburg, D., Ahonen, L. R., Chan, T., Fu, Y., Kim, C., Laurila, T. M., Zhou, Y., Dada, L., Sulo, J., Flagan, R. C., Kulmala, M., Petäjä, T., and Lehtipalo, K.: Overview of measurements and current instrumentation for 1–10 nm aerosol particle number size distributions, J Aerosol Sci, 148, 105584, https://doi.org/10.1016/j.jaerosci.2020.105584, 2020.

Kulmala, M., Kontkanen, J., Junninen, H., Lehtipalo, K., Manninen, H. E., Nieminen, T., Petaja, T., Sipila, M., Schobesberger, S., Rantala, P., Franchin, A., Jokinen, T., Jarvinen, E., Aijala, M., Kangasluoma, J., Hakala, J., Aalto, P. P., Paasonen, P., Mikkila, J., Vanhanen, J., Aalto, J., Hakola, H., Makkonen, U., Ruuskanen, T., Mauldin, R. L., Duplissy, J., Vehkamaki, H., Back, J., Kortelainen, A., Riipinen, I., Kurten, T., Johnston, M. V., Smith, J. N., Ehn, M., Mentel, T. F., Lehtinen, K. E. J., Laaksonen, A., Kerminen, V. M., and Worsnop, D. R.: Direct Observations of Atmospheric Aerosol Nucleation, Science, 339, 943-946, 10.1126/science.1227385, 2013.

Mikkonen, S., Németh, Z., Varga, V., Weidinger, T., Leinonen, V., Yli-Juuti, T., and Salma, I.: Decennial time trends and diurnal patterns of particle number concentrations in a Central European city between 2008 and 2018, Atmos. Chem. Phys. Discuss., 2020, 1-27, 10.5194/acp-2020-305, 2020.

Vana, M., Komsaare, K., Horrak, U., Mirme, S., Nieminen, T., Kontkanen, J., Manninen,

H. E., Petaja, T., Noe, S. M., and Kulmala, M.: Characteristics of new-particle formation at three SMEAR stations, Boreal Environ Res, 21, 345-362, 2016.

Zhou, Y., Dada, L., Liu, Y., Fu, Y., Kangasluoma, J., Chan, T., Yan, C., Chu, B., Daellenbach, K. R., Bianchi, F., Kokkonen, T. V., Liu, Y., Kujansuu, J., Kerminen, V. M., Petäjä, T., Wang, L., Jiang, J., and Kulmala, M.: Variation of size-segregated particle number concentrations in wintertime Beijing, Atmos. Chem. Phys., 20, 1201-1216, 10.5194/acp-20-1201-2020, 2020.

---

## Author Comment (AC1) · 14 Aug 2020

We would like to thank Referee 1 for valuable comments and suggestions. Our responses to this Referee's comments are stated below. Point-to-point response to each of the review comments is attached as given below. We highlighted the changed or modified part in revised manuscript using blue color for easily visible to editor. We believe this revised version is much better improve than the original version.

Please also note the supplement to this comment:
https://acp.copernicus.org/preprints/acp-2020-390/acp-2020-390-AC1-supplement.pdf

[Figure]

[Figure]

**Supplement:**

**Reviewer 1:**

The authors present beautiful data on NPF in the Arctic. They showed that the NPF events were correlated with DMS production and $NH_3$ concentration. I agree importance of those chemical components for NPF events in the Arctic region. The paper is well written and it can be accepted after some revisions and discussions. I have a comment on the precursor of the NPF event.

1) Is it possible to add analysis on the correlations between NPF and $SO_2$ gas concentration (and other possible precursors)?

Answer: The reviewer made a good point here. We appreciate for useful comments raised by this reviewer. More analysis on correlations among particle number and gas ($NH_3$, $SO_2$, and $H_2SO_4$) concentrations was conducted. We added the results for daily correlations between 1) $N_{3-25}$ and $SO_2$, 2) $N_{3-25}$ and $H_2SO_4$ derived from $SO_2$, temperature, RH, CS and solar radiation data, and 3) $N_{3-25}$ and $NH_3$ concentrations. It was found that the $SO_2$ and $NH_3$ were not significantly correlated with the $N_{3-25}$ (an increasing trend of $NH_3$ with the $N_{3-25}$ was observed but was not statistically significant). However, the $N_{3-25}$ was significantly correlated with the $H_2SO_4$ (r = 0.36), suggesting that the $H_2SO_4$ should play an important role in nucleation and growth. The results and discussion on this issue were added in manuscript as follows:

Page 11, line 329-336:
"The NH3 concentration was higher on NPF event days than on non-event days as shown in Figure 9 ($p$-value < 0.001), similar to results shown in Dall'Osto et al. (2017), although daily $NH_3$ concentration was not significantly correlated with the N3-25 as shown in Figure S5 in the Supplement. The $NH_3$ in the Arctic can originate from biological and animal sources (e.g., seabird colonies) (Tovar-Sánchez et al., 2010; Croft et al., 2016; Dall'Osto et al., 2017). The $SO_2$ was not significantly higher on NPF event days than on non-event days (Figure 9), and not significantly correlated with the $N_{3-25}$ (Figure S5 in the Supplement). On the other hand, the $H_2SO_4$ was found to be higher on the NPF event days (Figure 9) and was correlated with the $N_{3-25}$ (Figure S5 in the Supplement), suggesting that the $H_2SO_4$ should play an important role in nucleation and growth."

Revised Figure 9:

[Figure]

Figure 9. Comparison of average nss-$SO_4^{2-}$ ratio (nss-$SO_4^{2-}$/total $SO_4^{2-}$), $NH_3$, $SO_2$, and $H_2SO_4$ concentrations between NPF events and non-event days: error bar and stars represent the standard deviation and $p$-values of a t-test (ns: $> 0.05$, *: $\leq 0.05$, **: $\leq 0.01$, ***: $\leq 0.001$), respectively.

New Figure S5 in the Supplement:

[Figure]

(a)

[Figure]

(b)

(c)

Figure S5. Correlations of daily $N_{3-25}$ versus (a) daily $NH_3$, (b) daily $SO_2$, and (c) daily $H_2SO_4$ concentrations during the measurement period. The dashed line represents a linear regression line with a correlation coefficient (r).

[revised manuscript text omitted]

(a)

[Figure]

(b)

[Figure]

(c)

Figure S5. Correlations of daily $N_{3-25}$ versus (a) daily $NH_3$, (b) daily $SO_2$, and (c) daily $H_2SO_4$ concentrations during the measurement period. The dashed line represents a linear regression line with a correlation coefficient (r).

[Figure]

Figure S6. NPF event probability distribution with daily CS and temperature. The cell size was 2 K (temperature) and the ratio of 1.26 between two consecutive CS values.

Table S1. Average concentrations of ionic species ($Na^+$, $Mg^{2+}$, $K^+$, $NH_4^+$, $NO_3^-$, $SO_4^{2-}$, and $Cl^-$) in particulate matter and gaseous species ($NH_3$, $SO_2$, and $H_2SO_4$) in different seasons from 2016 to 2018.

| | Unit | Spring | Summer | Fall | Winter |
|---|---|---|---|---|---|
| $Na^+$ | $\mu g\ m^{-3}$ | 0.27±0.38 | 0.18±0.28 | 0.22±0.28 | 0.31±0.33 |
| $Mg^{2+}$ | $\mu g\ m^{-3}$ | 0.04±0.08 | 0.02±0.04 | 0.03±0.04 | 0.05±0.05 |
| $K^+$ | $\mu g\ m^{-3}$ | 0.05±0.07 | 0.03±0.02 | 0.03±0.02 | 0.03±0.02 |
| $NH_4^+$ | $\mu g\ N\ m^{-3}$ | 0.04±0.05 | 0.02±0.03 | 0.02±0.03 | 0.02±0.02 |
| $NO_3^-$ | $\mu g\ N\ m^{-3}$ | 0.02±0.02 | 0.02±0.02 | 0.02±0.04 | 0.02±0.02 |
| $SO_4^{2-}$ | $\mu g\ S\ m^{-3}$ | 0.19±0.18 | 0.08±0.10 | 0.08±0.09 | 0.11±0.20 |
| $Cl^-$ | $\mu g\ m^{-3}$ | 0.39±0.63 | 0.24±0.43 | 0.35±0.50 | 0.52±0.59 |
| $NH_3$ | $\mu g\ N\ m^{-3}$ | 0.13±0.60 | 0.16±0.22 | 0.10±0.10 | 0.08±0.07 |
| $SO_2$ | $\mu g\ S\ m^{-3}$ | 0.09±0.22 | 0.08±0.11 | 0.08±0.13 | 0.09±0.27 |
| $H_2SO_4$ | $10^5$ molecules $cm^{-3}$ | 7.43±8.16 | 8.59±8.64 | 5.52±8.91 | 0.95±0.69 |

---

## Author Comment (AC2) · 14 Aug 2020

We would like to Wolfgang Junkermann for valuable comments and suggestions. Our responses to this Referee's comments are stated below. Point-to-point response to each of the review comments is attached as given below. We highlighted the changed or modified part in revised manuscript using blue color for easily visible to editor. We believe this revised version is much better improve than the original version.

Please also note the supplement to this comment:
https://acp.copernicus.org/preprints/acp-2020-390/acp-2020-390-AC2-supplement.pdf

[Figure]

**Supplement:**

**Wolfgang Junkermann-short comments:**

Interesting paper, however, I have some comments regarding the origin of the particles and/or precursors. The authors claim, that due to the small size occasionally observed the particles have to be produced in the vicinity of the Zeppelin station. That's plausible. However, in that case it would be interesting whether anthropogenic emissions for example from the port may contribute. The frequency of particle events fits well into the frequency of ships in Ny-Ålesund. That port and ships affect the site is clearly shown for example in Eckhardt et al, 2013, here also the typical meteorological conditions for such an anthropogenic contamination at Zeppelin.

The event day May 14, 2018 is one of these days with low and variable winds at the Zeppelin site discussed in the Eckhardt paper (see also the PANGANEA meteorological data for this day). Also HYSPLIT, when calculated with the higher resolution of 0.5 degree instead of the default GDAS 1 degree shows an air mass trajectory sweeping over the Kongsfjord and the port of Ny-Ålesund while the 1 degree resolution does not show this Ny-Ålesund loop. A more detailed local meteorology would be very helpful in the search for the origin of nucleation mode particle precursors.

Second, the presence of cruise ships is roughly the same like the frequency of particle events during the summer months (May to August). These ships stay in the port normally for the daylight hours and leave the port at 16:00 local time. However, they are often in the fjord even for a longer time span. For the port there is the harbor keeping a record. For the whole archipelago the AIS database (www.marinetraffic.com) may be used. Ships are emitting a huge amount of ultrafine and even nucleation mode particles, especially when they are equipped with catalysts for NOx removal in emission control areas, see for example Kivekäs et al, 2014 or Kecorius et al, 2016. A single cruise ship in the vicinity (and up to 50 NM upwind) thus might be a dominating source of either primary nanoparticles or of nanoparticle precursors, especially in the otherwise pristine environment of Spitzbergen where normally sulphur compounds are thought to originate from DMS (open sea) and ammonia from seabird colonies (islands).

It would be good when such anthropogenic contamination could be excluded. Enclosed is the HYSPLIT analysis for May 14, 3 trajectories, two hour intervals

[Figure]

Fig. 1.

Answer: The reviewer made a good point here. Also, we would like to give many thanks to this reviewer for various useful comments. As suggested by the reviewer, the effects of anthropogenic sources (e.g., downtown, local port, and cruise ship) on the NPF were examined by using local wind and air mass trajectory data to find whether the air mass or wind passed over the Ny-Ålesund downtown or port before arriving our site. Also, the BC concentration (newly obtained from Zeppelin station), typically emitted from primary combustion sources, was used to exclude the effect of primary combustion sources on the NPF. We found that the air mass and wind passed over the downtown including the local port area during only two NPF events out of the whole NPF events (170 events). During these two NPF events, the BC concentration little increased. Thus, we believe the effect of anthropogenic sources on the observed NPF should be small. Also, we filtered out two NPF events with BC concentration increased when the wind direction coming from the Ny-Ålesund downtown or port. Thus, these two NPF were removed in our NPF data analysis. However, further studies may be required to examine NPF events caused by emissions from ship traffics.

We added discussion on this issue as given below.

Page 7, line 204-212:
"It was shown that the concentration of fine particles could be affected by local combustion sources such as local port and cruise ships (Eckhardt et al., 2013). The effects of anthropogenic sources (e.g., downtown, local port, and cruise ship) on the NPF were examined by using local wind and air mass trajectory data to find whether air mass or wind passed over the Ny-Ålesund downtown and local port during NPF events. Also, the concentration of black carbon (BC) at Zeppelin, typically emitted from primary combustion sources, was used to examine the effect of primary combustion sources on the NPF. We found that the air mass and wind passed over the downtown including the local port during only two NPF events out of whole NPF events (170 events). During these two NPF events, the BC concentration little increased. Thus, we believe the effect of anthropogenic sources on the NPF should be small. Also, in our NPF data analysis we filtered out two NPF events having increased BC concentration and wind direction coming from the Ny-Ålesund downtown or port."

Page 4, line 116-117:

"In addition, the hourly black carbon (BC) data at Zeppelin were used to examine the effect of primary combustion sources on the NPF."

We also added more detailed information for this reviewer.

Table.

| Cases | Number of NPF events |
|---|---|
| Air mass and wind passing over Ny-Ålesund downtown and local port area | 2 |
| Wind passing over Ny-Ålesund downtown and local port area with the increase of BC | 2 |
| No air mass and wind passing over Ny-Ålesund downtown and local port area | 166 |

[Figure]

Figure. Data for number size distribution, wind direction, BC concatenation, and $O_3$ concentration on May 14, 2018 – May 15, 2018 (NPF event). The BC concentration was not significantly enhanced during the NPF event. Also, the level of the BC was so low.

[revised manuscript text omitted]

(a)

[Figure]

(b)

[Figure]

Figure S5. Correlations of daily $N_{3-25}$ versus (a) daily $NH_3$, (b) daily $SO_2$, and (c) daily $H_2SO_4$ concentrations during the measurement period. The dashed line represents a linear regression line with a correlation coefficient (r).

[Figure]

Figure S6. NPF event probability distribution with daily CS and temperature. The cell size was 2 K (temperature) and the ratio of 1.26 between two consecutive CS values.

Table S1. Average concentrations of ionic species ($Na^+$, $Mg^{2+}$, $K^+$, $NH_4^+$, $NO_3^-$, $SO_4^{2-}$, and $Cl^-$) in particulate matter and gaseous species ($NH_3$, $SO_2$, and $H_2SO_4$) in different seasons from 2016 to 2018.

| | Unit | Spring | Summer | Fall | Winter |
|---|---|---|---|---|---|
| $Na^+$ | $\mu g\ m^{-3}$ | 0.27±0.38 | 0.18±0.28 | 0.22±0.28 | 0.31±0.33 |
| $Mg^{2+}$ | $\mu g\ m^{-3}$ | 0.04±0.08 | 0.02±0.04 | 0.03±0.04 | 0.05±0.05 |
| $K^+$ | $\mu g\ m^{-3}$ | 0.05±0.07 | 0.03±0.02 | 0.03±0.02 | 0.03±0.02 |
| $NH_4^+$ | $\mu g\ N\ m^{-3}$ | 0.04±0.05 | 0.02±0.03 | 0.02±0.03 | 0.02±0.02 |
| $NO_3^-$ | $\mu g\ N\ m^{-3}$ | 0.02±0.02 | 0.02±0.02 | 0.02±0.04 | 0.02±0.02 |
| $SO_4^{2-}$ | $\mu g\ S\ m^{-3}$ | 0.19±0.18 | 0.08±0.10 | 0.08±0.09 | 0.11±0.20 |
| $Cl^-$ | $\mu g\ m^{-3}$ | 0.39±0.63 | 0.24±0.43 | 0.35±0.50 | 0.52±0.59 |
| $NH_3$ | $\mu g\ N\ m^{-3}$ | 0.13±0.60 | 0.16±0.22 | 0.10±0.10 | 0.08±0.07 |
| $SO_2$ | $\mu g\ S\ m^{-3}$ | 0.09±0.22 | 0.08±0.11 | 0.08±0.13 | 0.09±0.27 |
| $H_2SO_4$ | $10^5$ molecules $cm^{-3}$ | 7.43±8.16 | 8.59±8.64 | 5.52±8.91 | 0.95±0.69 |

---

## Author Comment (AC3) · 14 Aug 2020

We would like to thank Referee 2 for valuable comments and suggestions. Our responses to this Referee's comments are stated below. Point-to-point response to each of the review comments is attached as given below. We highlighted the changed or modified part in revised manuscript using blue color for easily visible to editor. We believe this revised version is much better improve than the original version.

Please also note the supplement to this comment:
https://acp.copernicus.org/preprints/acp-2020-390/acp-2020-390-AC3-supplement.pdf

[Figure]

**Supplement:**

**Reviewer 2:**

"Atmospheric new particle formation characteristics in the Arctic as measured at Mount Zeppelin, Svalbard, from 2016 to 2018" by Haebum Lee et al.

General:

To assess the effect of ocean biology on atmospheric new particles formation observed between 2016-2018 at the Mount Zeppelin, Svalbard, this study was set to explore (using MODIS monthly mean satellite Chlorophyll-a concentrations) whether any direct link or correlation exist between ocean biological sources and the observed frequency of occurrence of nanoparticles as small as 3 nm diameter.

It was concluded that nanoparticles increased more frequently when the origin of air masses reaching the Zeppelin observatory overlapped with regions having strong Chlorophyll-a concentration and dimethyl sulfide (DMS) production capacity, and was also correlated with increased daily $NH_3$ concentrations. Moreover, the authors argue that the primary drivers of the observed new particles formation are the seasonal cycles of ocean biological activity or presence of sea bird colonies. However, it's essential to keep in mind that overlap or correlations do not prove a causal relationship. Unfortunately, the authors do not present any assessment of the reliability and credibility of the derived correlation.

Moreover, the literature survey is not fully convincing; there is some previous observational evidence on new particle formation from the Zeppelin observatory that reflect the seasonal cycle of gel-forming marine microorganisms and their controlling factors, that would seem to have an essential bearing on the results obtained and, thus, appear to merit discussion. Examples are Heintzenberg et al., 2017; Karl et al., 2019 and Mashayekhy Rad et al., 2019.

Furthermore, the satellite derived Chlorophyll-a numbers used in this study cannot be used in order to demonstrate a biological source in terms of biological activity (productivity/bloom) or phytoplankton primary productivity for the following reason; Chlorophylla is used a proxy of phytoplankton biomass whereas a phytoplankton bloom in various phases is the combined effect of phytoplankton production and zooplankton grazing, cell lysis and/or bacterial degradation. To be able to make any kind of statement relating the biological activity, phytoplankton biomass- and speciation, zooplankton etc. to DMS water concentrations, data on these parameters are required, covering the period 2016-2018.

The generally, over marine areas, poor correlation observed between parallel measurements of seawater DMS and phytoplankton biomass (Chlorophyll-a) has been explained as a consequence of the species specificity of DMS production followed upon the production of from extra-cellular (dimethyl sulfonium propionate) DMSPp.

A further very important reason for the in general poor correlation between DMS in the water and in algae biomass is that DMS is produced from DMSPd and not from intra-cellular DMSPp. This means that the production and breakdown of DMS in the water column must be looked on as a result of complex physiological and ecological interactions, as demonstrated in Leck et al. (1990). To be able to make any kind of statement linking the biological productivity, phytoplankton biomass and DMS water concentrations, data on the above-mentioned parameters are all required. As no such measurements were performed it is not valid to extrapolate results of high satellite derived Chlorophyll-a concentrations in the surface waters to DMS in the water column which will be subsequent emitted into the atmosphere and there undergo photochemical oxidation to form various intermediate products and, ultimately, sulphuric acid promoted nucleation.

Therefore, the statements causally linking the effect of ocean biology or as referred to in this study "DMS production capacity" on atmospheric new particles formation observed at the Zeppelin observatory are not well founded or supported and should be removed or de-emphasized.

As the conclusions are based on essential inputs of MODIS monthly mean satellite Chlorophyll-a concentrations and the suggested presence of bird colonies, which are both not constrained by measured in-situ data and that it seems that the authors have a somewhat limited conceptional understanding of the detailed processes involved needed for a successful assessment the effect of ocean biology on new particle formation observed at the Zeppelin observatory, I could only recommend the manuscript for publication after major revision according to the given comments and suggestions.

Although many of the above-identified parameters are not given as an integrated part of the present study, the manuscript has an important value as a descriptive data report of the seasonal life cycle of nanometer sized particles down to 3 nm diameter (with a relatively high temporal resolution), a parameter in general sparsely measured in the marine environment and specifically so in the Arctic.

Answer: The reviewer made a good point there. We appreciate for useful comments raised by this reviewer. We have done our best to address the reviewer's important questions. Also, conclusive statements were avoided through the modified manuscript, and more statistical analyses were included (i.e., new correlation analysis and t-test results were added). Detailed answers for questions and comments raised by the reviewer were given below.

We totally agreed with the reviewer. As suggested by the reviewer, the DMS production and biological activity should include complex physiological and ecological interactions. The DMS in the ocean is produced by complicate microbial food-web processes (Stefels et al., 2007). In general, sea surface DMS maximum occurs following local phytoplankton biomass maxima, thereby leading to lag periods on the order of several weeks to months (so called DMS summer paradox) (Galí and Simó, 2015). This phenomenon could be explained by several key processes: a succession in phytoplankton composition, grazing by zooplankton on DMSP-containing phytoplankton and the bacterial degradation of DMSP into DMS (Polimene et al., 2012). However, a clear temporal correlation between atmospheric (and/or seawater) DMS level and phytoplankton biomass (i.e., chlorophyll-α concentration) has been observed for the ocean domains where the strong DMS-producer (both containing high intra cellular DMSP content and DMSP cleavage enzyme) such as haptophytes and dinoflagellates are predominating (e.g., Arnold et al., 2010; Park et al., 2013; Park et al., 2018; Uhlig et al. 2019; Zhang et al., 2020). Only limited number of phytoplankton class including dinoflagellates and haptophytes possess enzyme that can convert DMSP into DMS during their growth (Alcolombri et al., 2015). In particular, *Emiliania huxleyi* and *Phaeocystis sp.* which are highly abundant haptophyte in high latitude oceans play key roles in controlling global DMS emission because the DMS production capacity of these species is much higher than other globally abundant phytoplankton species (Liss et al., 1994; McParland and Levine, 2019). For example, multi-year measurements of atmospheric DMS mixing ratios at Zeppelin station showed a strong correlation between sea-surface chlorophyll-α concentration (estimated by MODIS-aqua) and atmospheric DMS levels (Park et al., 2013; Park et al., 2018). Furthermore, relationships between the atmospheric DMS and phytoplankton biomass were regionally and temporally varied with the relative abundance of strong DMS(P)-producer (Park et al., 2018). This is because the oceanic DMS production in vicinity of the observation site (i.e., Greenland and Barents Seas) largely governed by direct DMS exudation of phytoplankton that has both high cellular DMSP content and DMSP-cleavage enzyme during phytoplankton bloom period. Recent study conducted at remote Antarctic site also revealed that the number concentration of nano-size particles (3-10 nm in diameter) was positively correlated with the chlorophyll-α concentration during the period when strong DMS-producer predominate (dominance of Phaeocystis >50%; estimated by PHYSAT algorithm) (Jang et al., 2019). Thus, we added the following discussion on this issue in the manuscript as given below.

Page 9-10, line 295-316:
"The DMS in the ocean is produced by complicate microbial food-web processes (Stefels et al., 2007). In general, sea surface DMS maximum occurs following local phytoplankton biomass maxima, thereby leading to lag periods on the order of several weeks to months (so called DMS summer paradox) (Galí and Simó, 2015). This phenomenon could be explained by several key processes: a succession in phytoplankton composition, grazing by zooplankton on DMSP-containing phytoplankton and the bacterial degradation of DMSP into DMS (Polimene et al., 2012). However, a clear temporal correlation between atmospheric (and/or seawater) DMS level and phytoplankton biomass (i.e., chlorophyll-α concentration) has been observed for the ocean domains where the strong DMS-producer (both containing high intra cellular DMSP content and DMSP cleavage enzyme) such as haptophytes and dinoflagellates are predominating (e.g., Arnold et al., 2010; Park et al., 2013; Park et al., 2018; Uhlig et al. 2019; Zhang et al., 2020). Only limited number of phytoplankton class including dinoflagellates and haptophytes possess enzyme that can convert DMSP into DMS during their growth (Alcolombri et al., 2015). In particular, *Emiliania huxleyi* and *Phaeocystis sp.* which are highly abundant haptophyte in high latitude oceans play key roles in controlling global DMS emission because the DMS production capacity of these species is much higher than other globally abundant phytoplankton species (Liss et al., 1994; McParland and Levine, 2019). For example, multi-year measurements of atmospheric DMS mixing ratios at Zeppelin station showed a strong correlation between sea-surface chlorophyll-α concentration (estimated by MODIS-aqua) and atmospheric DMS levels (Park et al., 2013; Park et al., 2018). Furthermore, relationships between the atmospheric DMS and phytoplankton biomass were regionally and temporally varied with the relative abundance of strong DMS(P)-producer (Park et al., 2018). This is because the oceanic DMS production in vicinity of the observation site (i.e., Greenland and Barents Seas) largely governed by direct DMS exudation of phytoplankton that has both high cellular DMSP content and DMSP-cleavage enzyme during phytoplankton bloom period. Recent study conducted at remote Antarctic site also revealed that the number concentration of nano-size particles (3–10 nm in diameter) was positively correlated with the chlorophyll-α concentration during the period when strong DMS-producer predominate (dominance of Phaeocystis > 50%; estimated by PHYSAT algorithm) (Jang et al., 2019)."

As suggested by the reviewers, we also added discussion with references (Heintzenberg et al., 2017; Karl et al., 2019; Mashayekhy Rad et al., 2019) to address that the gel-forming marine microorganisms could affect the NPF.

Page 7, line 223-226:
"In addition, it was suggested that fragmentation of primary marine polymer gels, which are derived from phytoplankton along the marginal ice zone, could be a source for atmospheric nanoparticles (NPF events below 10 nm) in the high Arctic boundary layer (Heintzenberg et al., 2017; Karl et al., 2019; Mashayekhy Rad et al., 2019)."

Detailed:
Page 2, line 50: relevant reference to add is Karl et al., 2013.
Answer: We added the reference "Karl et al., 2013".

Page 2, line 63: Please omit Covet et al., 1996.
Answer: We removed the reference "Covet et al., 1996".

Page 3, line 68: Please add Heintzenberg et al., 2017.
Answer: We added the reference "Heintzenberg et al., 2017".

Page 3, line 72: Please insert number between measured and size. To not confuse the reader be specific on that you discuss size distributions by number, throughout the manuscript.
Answer: As suggested by the reviewer, we added "number" throughout the manuscript.

Page 3, line 78: Please remove the superscript (10) on "occur".
Answer: We removed the superscript (10) on "occur".

Page 3, line 80: Please motivate why you will use satellite-derived Chlorophyll-a concentration data to detect potential source regions for new particle formation. Please also add that Chlorophyll-a is a proxy of phytoplankton biomass only.

Answer: As suggested by the reviewer, the following statements were added.

Page 3, line 80-83:
"The chlorophyll-α which is involved in oxygenic photosynthesis in ocean has been considered as a proxy for phytoplankton biomass only. Recent studies showed that there was a strong correlation between sea-surface chlorophyll-α concentration (estimated by MODIS-aqua) and atmospheric DMS levels at Zeppelin station (Park et al., 2013; Park et al., 2018)."

Page 3, lines 90-93: Observations are a challenge and specifically so in the pristine remote Arctic marine environment. Specific to the Zeppelin observatory extreme care must be exercised to prevent interference from local pollution (ship traffic) and thus contamination of the air samples. Please add information on the implemented procedure to detect and avoid contamination by local pollution or from long-range transport (southerly air mass origin). Did you have any automatically interruption of the sampling when necessary, due to unfavorable conditions (pollution sensor)?

Answer: The reviewer made a good point here. This is the similar question to one raised by the reviewer 2. As suggested by the reviewer, the effects of anthropogenic sources (e.g., downtown, local port, and cruise ship) on the NPF were examined by using local wind and air mass trajectory data to find whether the air mass or wind passed over the Ny-Ålesund downtown or port before arriving our site. Also, the BC concentration (newly obtained from Zeppelin station), typically emitted from primary combustion sources, was used to exclude the effect of primary combustion sources on the NPF. We found that the air mass and wind passed over the downtown including the local port area during only two NPF events out of the whole NPF events (170 events). During these two NPF events, the BC concentration little increased. Thus, we believe the effect of anthropogenic sources on the observed NPF should be small. Also, we filtered out two NPF events with BC concentration increased when the wind direction coming from the Ny-Ålesund downtown or port. Thus, these two NPF were removed in our NPF data analysis. However, further studies may be required to examine NPF events caused by emissions from ship traffics.

We added discussion on this issue as given below.

Page 7, line 204-212:
"It was shown that the concentration of fine particles could be affected by local combustion sources such as local port and cruise ships (Eckhardt et al., 2013). The effects of anthropogenic sources (e.g., downtown, local port, and cruise ship) on the NPF were examined by using local wind and air mass trajectory data to find whether air mass or wind passed over the Ny-Ålesund downtown and local port during NPF events. Also, the concentration of black carbon (BC) at Zeppelin, typically emitted from primary combustion sources, was used to examine the effect of primary combustion sources on the NPF. We found that the air mass and wind passed over the downtown including the local port during only two NPF events out of whole NPF events (170 events). During these two NPF events, the BC concentration little increased. Thus, we believe the effect of anthropogenic sources on the NPF should be small. Also, in our NPF data analysis we filtered out two NPF events having increased BC concentration and wind direction coming from the Ny-Ålesund downtown or port."

Page 4, line 97: General to the manuscript. Please replace specie(s) with compounds or constituents as specie(s) belongs to the family of living organisms.
Page 4, line 97: All Ionic – and molecular formulas should be defined.

Answer: The molecular formula cannot be determined in this study because the concentration of each ionic species ($Na^+$, $Mg^{2+}$, $K^+$, $NH_4^+$, $NO_3^-$, $SO_4^{2-}$, and $Cl^-$) in bulk PM sample was measured.

Page 4, lines 97-101: Please give the 50% cut-off equivalent aerodynamic diameters (EADs) of the 3-stage filterpack sampler (type?). Also give details on the analytical methods used for both the particular matter and gas phase compounds collected. How were the blank levels determined? Analytical detection limits obtained for the various ions? Were any Quality checks of the IC-analyses performed? LOD, precision?

Answer: We added more detailed information on analytical methods as given below.

Page 4, line 101-108:
"Daily ionic species and gas data are daily measurements collected with a 3-stage filterpack sampler (NILU prototype) with no pre-impactor. The size cut off of the inlet section is approximately 10 µm. Field blanks were prepared in the same as the other samples. It should be noted that for the nitrogen compounds the separation of gas and aerosol might be biased due to the volatile nature of $NH_4NO_3$. The detection limits were 0.05 µg N $m^{-3}$ and 0.01 µg S $m^{-3}$ for $NH_3$ and $SO_2$, respectively, and 0.01 µg $m^{-3}$ for $Na^+$, $Mg^{2+}$, $K^+$, $NH_4^+$, and $Cl^-$, 0.01 µg N $m^{-3}$ for $NO_3^-$, and 0.01 µg S $m^{-3}$ for $SO_4^{2-}$. The data quality management and system are accredited in accordance to NS-EN ISO / IEC 1702 standards. The detailed information of sampling method and analysis can be found elsewhere (EMAP 2014; Aas et al., 2019)."

Page 4, line 102: Please define AWIPEV.

Answer: We added the full name of "AWIPEV".
"… at the AWIPEV (the Alfred Wegener Institute Helmholtz Centre for Polar and Marine Research and the French Polar Institute Paul Emile Victor)…"

Page 4, line 106: Please define EMEP, ACTRIS, GAW-WDCA.
Answer: We added the full name of "EMPE", "ACTRIS", and "GAW-WDCA".
"… monitoring programmes (i.e., EMEP (European Monitoring and Evaluation Programme), ACTRIS (Aerosols, Clouds and Trace gases Research InfraStructure Network), GAW-WDCA (Global Atmospheric Watch - the World Data Centre for Aerosols))"

Page 5, line 142: Please add details on how the sulfuric acid number concentration was predicted from the measured daily $SO_2$, please also discuss the quality of the data in use.

Answer: As suggested by the reviewer, we added more discussion on how to estimate $H_2SO_4$ concentration.

Page 5, line 151-161:
"The $H_2SO_4$ molecular concentration was predicted from the measured daily $SO_2$, hourly CS, hourly solar radiation, and hourly meteorological data (RH and temperature) using the method proposed by Mikkonen et al. (2011). The empirical proxy model of $H_2SO_4$ is given by:

$$[H_2SO_4] = a \cdot k \cdot [SO_2]^b \cdot SRAD^c \cdot (CS \cdot RH)^d \qquad (4)$$

where $[SO_2]$ is the $SO_2$ molecular concentration (molecules cm$^{-3}$), SRAD is the solar radiation (W m$^{-2}$), CS is the condensation sink (s$^{-1}$), RH is the relative humidity (%), and k is the reaction rate constant depending on ambient temperature (see detailed definition for k in Eq. (3) of Mikkonen et al., 2011) with coefficients of a = $8.21 \times 10^{-3}$, b = 0.62, c = 1, and d = -0.13. The $H_2SO_4$ concentration at Zeppelin was $5.98 \times 10^4$–$3.19 \times 10^6$ molecules cm$^{-3}$ during the summer in 2008 (Giamarelou et al., 2016) which is in a similar range to ours ($2.69 \times 10^4$–$7.68 \times 10^6$ molecules cm$^{-3}$) in 2016-2018."

Page 5, line 154: Be specific, which are the precursor gases in mind? How do you verify their abundance? How will a proxy measure of marine phytoplankton biomass (Chlorophyll-a) influence the availability of atmospheric precursor gases for new particle formation?

Answer: We specified possible candidate precursors in that statements.

Page 6, line 173-176:
"In addition, marine biogenic sources, which provide gaseous precursors (e.g., DMS, $H_2SO_4$, and $NH_3$) for nanoparticle formation, were known as abundant in summer. It was observed that the percentage of air mass passing over high chlorophyll-α (MODIS data) region, and $H_2SO_4$ and $NH_3$ concentrations measured at the site increased in summer (Figure S2 and Table S1 in the Supplement)."

Page 6, line 176: Please add the results by Heintzenberg et al., 2017.

Answer: We added the results by "Heintzenberg et al. (2017)".

Page 7, line 196-200:
"The mean occurrence percentage of NPF days (all types) per year from 2016 to 2018 was 23%. Dall'Osto et al. (2017) found that the average of yearly NPF occurrence from 2000 to 2010 was 18%, lower than our value, and that this increased over time as the coverage of sea-ice melt increased. Based on Heintzenberg et al. (2017) study, the mean occurrence percentage of NPF days per year from 2006 to 2015 was estimated to be around 20%.

Page 6, line 178: Please clarify which results that support the following statement "In addition, DMS originating from marine sources can be a key precursor contributing to NPF in the remote marine atmosphere".
Answer: We added the following references to support the statement.

Page 7, line 200-201:
"In addition, DMS originating from marine sources can be a key precursor contributing to NPF in the remote marine atmosphere (Leaitch et al., 2013; Park et al., 2017; Jang et al., 2019)."

Page 6, lines 181-182; Page 7, lines 203-204 : How do you explain the poor correlation observed between the highest percentage of new particle formation (Fig. 5), and the relatively low MODIS monthly mean satellite-derived phytoplankton biomass (Chlorophyll-a) concentrations in the month of August (Fig. S1 bottom and Fig.6)?

Answer: The reviewer made a good point here. The figure led to confusion to readers. To address this issue, we added new results for "air mass exposure to chlorophyll-$\alpha$" ($E_{chl}$) which reflects the influences of phytoplankton biomass and atmospheric DMS oxidation on the DMS mixing ratio of the air mass arriving at Zeppelin (Park et al., 2018) as shown in Figure S2 in Supplement. It provides the measure of potential DMS production capacity of the ocean air mass passed over. The "air mass exposure to chlorophyll-$\alpha$" ($E_{chl}$) was calculated by (Park et al., 2018):

$$E_{chl} = \frac{\sum_{t=1}^{48} Chl_t \cdot \exp\left(-\alpha\left(\frac{t}{24}\right)\right)}{n}$$

where "$Chl_t$" is the 8-day mean chlorophyll-$\alpha$ concentration within a radius of 25 km at given time point (t = 1 to 48) along the air mass back-trajectory, and "n" is the total number of time points for which valid chlorophyll values are available. The term $\exp(-\alpha(t/24))$ corresponds to the normalization of the photo-decay, where "$\alpha$" is the decay constant of DMS in the atmosphere due to photochemical processes. A value of 0.43 was used for $\alpha$.

It was found that "air mass exposure to chlorophyll-$\alpha$" ($E_{chl}$) was correlated well (r = 0.69) with the NPF occurrence frequency as shown in Figure S2 in the Supplement, compared to the average chlorophyll-$\alpha$ concentration over the area (70-85ºN, 25ºW-50ºE). We added this discussion in the followings.

Page 8, line 240-249:

"Figure 6 shows the MODIS monthly chlorophyll-α concentrations around Svalbard, which increased from April and decreased after August, consistent with the NPF occurrence frequency. The chlorophyll-α concentration was intense in the ocean regions southwest and southeast of Svalbard. A recent study revealed that the DMS production capacity of the Greenland Sea (to the southwest) was 3 times greater than that of the Barents Sea (to the southeast) (Park et al., 2018); this is further discussed in the context of air mass trajectory data in a later section. Full monthly values of average chlorophyll-α concentration over the area (70-85ºN, 25ºW-50ºE) and "air mass exposure to chlorophyll-α" ($E_{chl}$) which explains the DMS mixing ratio of the air mass arriving at Zeppelin (Park et al., 2018) are summarized in Figure S2 in the Supplement. The $E_{chl}$ provides the measure of potential DMS production capacity of the ocean air mass passed over (Park et al., 2018). It was found that "air mass exposure to chlorophyll-α" ($E_{chl}$) was correlated well (r = 0.69 and *p*-value < 0.05; not shown) with the NPF occurrence frequency, compared to the average chlorophyll-α concentration over the area (70-85ºN, 25ºW-50ºE)."

[Figure]

Figure S2. Monthly values of average chlorophyll-α concentration over the area (70-85ºN, 25ºW-50ºE) and "air mass exposure to chlorophyll-α" ($E_{chl}$) calculated by Eq. (1) in Park et al. (2018) from March to September 2016 to 2018.

Page 6, line 187: I cite "In addition, the melting of sea ice in summer can increase the availability of marine biogenic sources, promoting NPF". Please specify which the sources you have in mind? Here I find the literature survey unconvincing; there is quite a large amount of previous observational evidence that would seem to have an essential bearing on the results obtained and, thus, appear to merit discussion. In this respect, it seems somewhat surprising that even not mention of or learn from the previous work by Leck and her colleagues over the last three decades on releases of atmospheric sulfur compounds and marine sea-spray aerosols (organic polymer gels/inorganic) over the Arctic pack ice area (incl. the marginal ice zone) in summer.

Answer: As suggested by the reviewers, we also added discussion with references (Heintzenberg et al., 2017; Karl et al., 2019; Mashayekhy Rad et al., 2019) to address that the gel-forming marine microorganisms could affect the NPF.

Page 7, line 223-226:
"In addition, it was suggested that fragmentation of primary marine polymer gels, which are derived from phytoplankton along the marginal ice zone, could be a source for atmospheric nanoparticles (NPF events below 10 nm) in the high Arctic boundary layer (Heintzenberg et al., 2017; Karl et al., 2019; Mashayekhy Rad et al., 2019)."

Page 7, lines 204-207; Page 8 lines 239-242; Page 9, line 281: Figure 6 shows that the Chlorophyll-concentrations were most pronounced in the ocean areas southwest and southeast of Svalbard. After the authors explored the potential source regions of the air masses in relation to occurrence of nanoparticles, it was found that increasing numbers of the latter occurred more frequently when the air trajectories passed over the oceanic regions to southwest and south of Svalbard but surprisingly not when passing over the ocean areas south east of Svalbard. This was explained by that the DMS production capacity of the southwest ocean was 3 times greater than that of the southeast ocean. To support the findings the authors used the results derived from a study based on data sets obtained between May and April in 2010, 2014 and 2015 by Park et al. (2018). Please give a detailed explanation to what "DMS production capacity" stands for and what it critically depends on. Please also explain how the findings by Park et al., 2018 covering only two months (April and May) of the biological season (this study April to October) and in different years, could be used to explain the findings in this study.

Answer: As suggested by the reviewer, we added the definition of the DMS production capacity and several references supporting the strong DMS production in the southwest ocean.

Page 9, line 290-294:
"… regions, and the DMS production capacity of the southwest ocean was 3 times greater than that of the southeast ocean. The DMS production capacity was defined as the potential amount of DMS produced from the phytoplankton biomass (Park et al., 2018). Several previous studies also support the strong DMS production capacity in the southwest ocean (Degerlund and Eilertsen, 2010; Galí and Simó, 2010). These results suggest that marine biogenic sources from the southwest ocean (Greenland Sea) region can play an important role in NPF in the Arctic."

Page 7, line 208: Please support your statement: "The existence of significant amounts of nanoparticles as small as 3 nm during NPF events at the study site suggests that NPF occurred there, rather than the particles being transported from other regions after growth." What is the expected atmospheric residence time of the nucleated particles?

Answer: The lifetime of the 3 nm particles in this study (by growth to particles larger than 7 nm) was estimated to be 2-3 hours on average. We added the following statement to explain this issue.

Page 9, line 271-276:
"The existence of significant amounts of nanoparticles as small as 3 nm during NPF events at the study site suggests that NPF occurred there, rather than the particles being transported from other regions after growth. In other words, if NPF occurred at other locations far from the study site, the nanoparticles would have grown during transport to the site and few 3 nm particles would have been detected there. The lifetime of the 3 nm particles in this study (growth to particles larger than 7 nm) was estimated to be 2-3 hours on average. It was reported that nanoparticles ($<$ 5 nm) in the troposphere could survive for several hours or less (Anastasio and Martin, 2001)."

Page 8, line 252: Please clarify the meaning of that non-sea-salt sulfate could have a secondary origin from oceanic DMS. Which formula was used in the calculations of non-sea-salt sulfate. How do you estimate the contribution from non-biogenic DMS sources? A much more strait forward comparison would be to use particulate methane sulfonate (MSA) concentrations of the total suspended particle samples.

Answer: We added the following statements to explain how to calculate the non-sea-salt sulfate.

Page 10-11, line 324-328:
"The non-sea salt sulfate ($nss\text{-}SO_4^{2-}$) could have had a secondary origin from the DMS from the sea (Park et al., 2017; Kecorius et al., 2019). The $SO_4^{2-}$ could also come from sea salt particles (primary production of $SO_4^{2-}$) (Karl et al., 2019). Thus, the concentration of $nss\text{-}SO_4^{2-}$ was derived from $nss\text{-}SO_4^{2-}$ ($\mu g\ m^{-3}$) = total $SO_4^{2-}$ ($\mu g\ m^{-3}$) – 0.252×$Na^+$ ($\mu g\ m^{-3}$) by using the measured $SO_4^{2-}$ and $Na^+$ concentrations (Zhan et al., 2017)."

Page 8, line 252: Could you please discuss how realistic your assumption on a DMS derived (sulfuric acid) nucleation mechanism is in respect to the findings by Pirjola et al. (2000), which showed that, under typical conditions in the MBL, homogeneous binary $H_2SO_4\text{-}H_2O$ nucleation will not occur and ternary $H_2SO_4\text{-}H_2O\text{-}NH_3$ nucleation will only be sufficiently effective to produce observable particles for DMS concentrations in the range of 400 ppt(v) or higher and very low aerosol condensation sinks.

Answer: We examined the importance of the DMS and other parameters on the occurrence of NPF. Our data are limited to fully explain the nucleation mechanism which is out of scope in this study. Further studies should be required to elucidate the nucleation mechanism by directly measuring chemical composition of nanoparticles and various precursor vapors. We added the following statement about this issue.

Page 11, line 336-337:

"Our data were limited to fully explain the nucleation mechanism. Further studies should be required to elucidate the nucleation mechanism by directly measuring chemical composition of nanoparticles and various precursor vapors."

Page 18, Figure 2: Please remove the superscript (37) on "SRAD".

Answer: We removed the superscript (37) on "SRAD".

[revised manuscript text omitted]

(a)

[Figure]

(b)

[Figure]

(c)

Figure S5. Correlations of daily $N_{3-25}$ versus (a) daily $NH_3$, (b) daily $SO_2$, and (c) daily $H_2SO_4$ concentrations during the measurement period. The dashed line represents a linear regression line with a correlation coefficient (r).

[Figure]

Figure S6. NPF event probability distribution with daily CS and temperature. The cell size was 2 K (temperature) and the ratio of 1.26 between two consecutive CS values.

Table S1. Average concentrations of ionic species ($Na^+$, $Mg^{2+}$, $K^+$, $NH_4^+$, $NO_3^-$, $SO_4^{2-}$, and $Cl^-$) in particulate matter and gaseous species ($NH_3$, $SO_2$, and $H_2SO_4$) in different seasons from 2016 to 2018.

| | Unit | Spring | Summer | Fall | Winter |
|---|---|---|---|---|---|
| $Na^+$ | $\mu g\ m^{-3}$ | 0.27±0.38 | 0.18±0.28 | 0.22±0.28 | 0.31±0.33 |
| $Mg^{2+}$ | $\mu g\ m^{-3}$ | 0.04±0.08 | 0.02±0.04 | 0.03±0.04 | 0.05±0.05 |
| $K^+$ | $\mu g\ m^{-3}$ | 0.05±0.07 | 0.03±0.02 | 0.03±0.02 | 0.03±0.02 |
| $NH_4^+$ | $\mu g\ N\ m^{-3}$ | 0.04±0.05 | 0.02±0.03 | 0.02±0.03 | 0.02±0.02 |
| $NO_3^-$ | $\mu g\ N\ m^{-3}$ | 0.02±0.02 | 0.02±0.02 | 0.02±0.04 | 0.02±0.02 |
| $SO_4^{2-}$ | $\mu g\ S\ m^{-3}$ | 0.19±0.18 | 0.08±0.10 | 0.08±0.09 | 0.11±0.20 |
| $Cl^-$ | $\mu g\ m^{-3}$ | 0.39±0.63 | 0.24±0.43 | 0.35±0.50 | 0.52±0.59 |
| $NH_3$ | $\mu g\ N\ m^{-3}$ | 0.13±0.60 | 0.16±0.22 | 0.10±0.10 | 0.08±0.07 |
| $SO_2$ | $\mu g\ S\ m^{-3}$ | 0.09±0.22 | 0.08±0.11 | 0.08±0.13 | 0.09±0.27 |
| $H_2SO_4$ | $10^5$ molecules $cm^{-3}$ | 7.43±8.16 | 8.59±8.64 | 5.52±8.91 | 0.95±0.69 |

---

## Author Comment (AC4) · 14 Aug 2020

We would like to thank Referee 3 for valuable comments and suggestions. Our responses to this Referee's comments are stated below. Point-to-point response to each of the review comments is attached as given below. We highlighted the changed or modified part in revised manuscript using blue color for easily visible to editor. We believe this revised version is much better improve than the original version.

Please also note the supplement to this comment:
https://acp.copernicus.org/preprints/acp-2020-390/acp-2020-390-AC4-supplement.pdf

[Figure]

**Supplement:**

**Reviewer 3:**

Review of 'Atmospheric new particle formation characteristics in the Arctic as measured at Mount Zeppelin, Svalbard, from 2016 to 2018' by Lee et al.

The manuscript studies the characteristics of NPF at Mount Zeppelin, a location in the Arctic far from direct anthropogenic emissions. The study compromises ~2 years of comprehensive valuable data suitable for NPF study. While NPF has been studied at the same location, the new data included in this study contains high time resolution of particle number size distributions of particle sizes relevant for new particle formation. The manuscript is well written, the methods used are clearly described and the literature review is thorough. I suggest publication in ACP after addressing the comments below.

General comments:
1. The exact dates of the measurements need to be reported to identify the reoccurrence of the NPF seasons. The authors mention '89% during the 27 months sampling period', but the exact months need to be mentioned.

Answer: As suggested by the reviewer, we added the detailed sampling period in the manuscript.

Page 6, line 163-164:
"The data coverage for the size distribution data collected by nano-SMPS was about 89% during the 27 months sampling period (Oct 2016 to Dec 2018)."

2. I agree with Anonymous Referee #1 on the necessity of showing the correlation between concentrations of precursor vapours and particle formation rates and growth rates. How do the concentrations of these vary between event days and nonevent days. Something like your figure 9 would be nice to show also for gas-phase precursors. It could be divided into monthly event days and non-event days.

Answer: The reviewer made a good point here. We appreciate for useful comments raised by this reviewer. More analysis on correlations among particle number and gas ($NH_3$, $SO_2$, and $H_2SO_4$) concentrations was conducted. We added the results for daily correlations between 1) $N_{3-25}$ and $SO_2$, 2) $N_{3-25}$ and $H_2SO_4$ derived from $SO_2$, temperature, RH, CS and solar radiation data, and 3) $N_{3-25}$ and $NH_3$ concentrations. It was found that the $SO_2$ and $NH_3$ were not significantly correlated with the $N_{3-25}$ (an increasing trend of $NH_3$ with the $N_{3-25}$ was observed but was not statistically significant). However, the $N_{3-25}$ was significantly correlated with the $H_2SO_4$ ($r = 0.36$), suggesting that the $H_2SO_4$ should play an important role in nucleation and growth. The results and discussion on this issue were added in manuscript as follows:

Page 11, line 329-336:

"The NH$_3$ concentration was higher on NPF event days than on non-event days as shown in Figure 9 ($p$-value < 0.001), similar to results shown in Dall'Osto et al. (2017), although daily NH$_3$ concentration was not significantly correlated with the N3-25 as shown in Figure S5 in the Supplement. The NH$_3$ in the Arctic can originate from biological and animal sources (e.g., seabird colonies) (Tovar-Sánchez et al., 2010; Croft et al., 2016; Dall'Osto et al., 2017). The SO$_2$ was not significantly higher on NPF event days than on non-event days (Figure 9), and not significantly correlated with the N$_{3-25}$ (Figure S5 in the Supplement). On the other hand, the H$_2$SO$_4$ was found to be higher on the NPF event days (Figure 9) and was correlated with the N$_{3-25}$ (Figure S5 in the Supplement), suggesting that the H$_2$SO$_4$ should play an important role in nucleation and growth."

Revised Figure 9:

[Figure]

Figure 9. Comparison of average nss-SO$_4^{2-}$ ratio (nss-SO$_4^{2-}$/total SO$_4^{2-}$), NH$_3$, SO$_2$, and H$_2$SO$_4$ concentrations between NPF events and non-event days: error bar and stars represent the standard deviation and $p$-values of a t-test (ns: > 0.05, *: ≤ 0.05, **: ≤ 0.01, ***: ≤ 0.001), respectively.

New Figure S5 in the Supplement:

[Figure]

(a)

[Figure]

(b)

[Figure]

(c)

Figure S5. Correlations of daily $N_{3-25}$ versus (a) daily $NH_3$, (b) daily $SO_2$, and (c) daily $H_2SO_4$ concentrations during the measurement period. The dashed line represents a linear regression line with a correlation coefficient (r).

3. The trends of the precursor vapors during the measurement period (sulfuric acid and ammonia), the number concentrations in different clusters, and different modes (3-7 nm, 7-25 nm) as well as the particle formation and growth rates can be shown as daily or weekly medians, maybe in the supplementary. Similar to Kalivitis et al. (2019) figures 8c and 8d or Mikkonen et al. (2020) figure 2.

Answer: We added such information in Figure S4 in the Supplement.

Page 9, line 268-270:
"Time series of daily GR and J in different modes ($GR_{3-7}$ and $J_{3-7}$, and $GR_{7-25}$ and $J_{7-25}$), weekly $N_{3-7}$ and $N_{7-25}$, and weekly $NH_3$ and $H_2SO_4$ are shown in Figure S4 in the Supplement."

New Figure S4 in the Supplement:

[Figure]

Figure S4. Time series of (a) weekly $N_{3-7}$, $N_{7-25}$, $NH_3$, and $H_2SO_4$, (b) daily GR and (c) daily J in different modes ($J_{3-7}$, $J_{7-25}$, $GR_{3-7}$, and $GR_{7-25}$).

4. You calculate $J_{3-7}$ but $GR_{3-25}$, although the GR is not constant over the size bin 3-25 nm (Kulmala et al., 2013). Calculating a size segregated GR, i.e. $GR_{3-7}$ and $GR_{7-25}$ is recommended especially looking at your figure 4 (upper left), the GR is not constant over these sizes.

Answer: The reviewer made a good point. As suggested by the reviewer, $GR_{3-7}$ and $GR_{7-25}$, and $J_{3-7}$ and $J_{7-25}$ were re-calculated, and related Figures were revised.

Page 8-9, line 250-270:

"To determine the characteristics of particle growth, we calculated the GR in the 3–7 nm, 7–25 nm, and 3–25 nm size ranges (i.e., $GR_{3-7}$, $GR_{7-25}$, and $GR_{3-25}$) for NPF events (Figure 7). The average $GR_{3-25}$ for all months was 2.66 nm h$^{-1}$, comparable to previously reported GR data (0.2–4.1 nm h$^{-1}$) in the Arctic region (Kerminen et al., 2018). The highest monthly average $GR_{3-25}$ was observed in July (3.03 nm h$^{-1}$) and the maximum individual value (6.54 nm h$^{-1}$) occurred in June. The averages of $GR_{3-7}$ and $GR_{7-25}$ were 2.07 nm h$^{-1}$ and 2.85 nm h$^{-1}$, respectively. However, the GR was much lower than the values observed in typical urban areas (Table 1), suggesting a lower availability of condensing vapors contributing to particle growth in the Arctic atmosphere. The formation rates of particles in the same size range as calculated GR were also derived. The averages of $J_{3-7}$, $J_{7-25}$, and $J_{3-25}$ during NPF events were 0.04 cm$^{-3}$ s$^{-1}$, 0.09 cm$^{-3}$ s$^{-1}$ and 0.12 cm$^{-3}$ s$^{-1}$, respectively. The highest monthly average and maximum for $J_{3-7}$ were both found in June, but for $J_{7-25}$ and $J_{3-25}$ were found in July. The formation rates (relative standard deviation (RSD) = 39–44%) varied by month more significantly than for GR (RSD = 27–33%). The formation rates in this study were much lower than those reported in continental areas (Stanier et al., 2004; Hamed et al., 2007; Wu et al., 2007; Manninen et al., 2010; Xiao et al., 2015; Shen et al., 2016; Cai et al., 2017). A good linear relationship was found between $J_{3-7}$ and $N_{3-7}$ (r = 0.97 and *p*-value < 0.001) as shown in Figure S3 in the Supplement, indicating that 3–7 nm particles were produced by gas-to-particle conversion rather than direct emissions in the particle phase (i.e., not primary) (Kalivitis et al., 2019). No significant correlation was found between $J_{3-7}$ and $GR_{3-7}$, suggesting that the vapors participating in the early stage of NPF could be at least partly different from the vapors contributing to subsequent particle growth (Nieminen et al., 2014). However, detailed chemical data for nanoparticles during formation and growth should be obtained to achieve complete understanding of the participating chemical species. Our data indicate that, although NPF occurrence frequency in the Arctic was comparable to continental areas, the J and GR were much lower. Time series of daily GR and J in different modes ($GR_{3-7}$ and $J_{3-7}$, and $GR_{7-25}$ and $J_{7-25}$), weekly $N_{3-7}$ and $N_{7-25}$, and weekly $NH_3$ and $H_2SO_4$ are shown in Figure S4 in the Supplement."

New Figure S3 in the Supplement:

[Figure]

Figure S3. Relationship between $N_{3-7}$ and $J_{3-7}$ during NPF events with a liner regression line and a correction coefficient (r).

5. I don't understand why you chose to present the data in UTC and not Local time. When using UTC, there is no relevance to solar radiation or to other locations. Please show your figures in local time (Figure 4 and Figure 5-middle). You can also show figure 5-middle relative to sunrise. See for example figure 6 in Dada et al. (2018).

Answer: As suggested by the reviewer, the UTC time was changed into the local time as shown in Figures 4 and 5.

6. What about nighttime clustering? your figure middle panel shows that the start time of NPF is around 20 UTC? also unit of time needs to be added to the figure or caption.

Answer: The unit of time was added in the figure and caption. The nighttime NPF also occurred in late fall to winter (20% out of total NPF events). The exact mechanism for this NPF was unclear to us. Nanoparticles formed at earlier times (daytime) in other places may be transported to the site during nighttime (Vehkamaki et al., 2004; Park et al., 2020). The discussion was added in manuscript as follows:

Page 8, Line 237-239:
"The nighttime NPF also occurred in late fall to winter (20% out of total NPF events). The exact mechanism for this NPF was unclear. Nanoparticles formed at earlier times (daytime) in other places may be transported to the site during nighttime (Vehkamaki et al., 2004; Park et al., 2020)."

7-1. There seems to be an effect of temperature as well as CS on the probability of NPF. See figure 13 in Dada et al. (2017).

Answer: As suggested by the reviewer, we discussed the effect of temperature on the NPF probability. A similar figure to the Dada et al. (2017) was also added in the Supplement (Figure S6).

Page 11, line 338-341:
"The NPF event probability distribution with daily CS and temperature was included in Figure S6 in the Supplement. The NPF event probability was calculated by the ratio of the NPF event days per total days for the given CS and temperature. The NPF event probability increased at moderate temperatures when the CS was low, while when the CS was high, the probability increased at relatively high temperature as shown in Figure S6 in the Supplement."

New Figure S6 in the Supplement:

[Figure]

Figure S6. NPF event probability distribution with daily CS and temperature. The cell size was 2 K (temperature) by the ratio of 1.26 between two consecutive CS values.

7-2. Does the occurrence of Arctic haze inhibit 3 nm clustering and growth?

Answer: The NPF still occurred during the Arctic haze period (April-May), but the NPF occurrence frequency was lower than summer. Refer to our previous statements as given below.

Page 7, line 216-223:

"Our results showed that NPF occurrence increased significantly in April, was maintained at a high level from May to August, then decreased in September and October. The average values of CS during NPF event and non-event days were $0.57\times10^{-3}$ s$^{-1}$ and $0.69\times10^{-3}$ s$^{-1}$, respectively. The higher biological and photochemical activity, lower transport of pollutants from mid-latitudes, and increased wet scavenging of particles (low CS) in summer likely favored NPF (Ström et al., 2009). In addition, the melting of sea ice in summer can increase the availability of marine biogenic sources, promoting NPF (Quinn et al., 2008; Tovar-Sánchez et al., 2010; Dall'Osto et al., 2018). Overall, NPF occurrence is mainly affected by the availability of solar radiation (photochemistry) and gaseous precursors in addition to the survival probability of clusters or particles (Kulmala et al., 2017)."

7-3. How different is the CS between NPF event days and non-event days? If possible, you could examine how CS varies between the air mass clusters.

Answer: As suggested by the reviewer, we compared CS during NPF event and non-event days. Also, information on air mass-dependent CS was added.

Page 7, line 218-219:
"The average values of CS during NPF event and non-event days were $0.57\times10^{-3}$ s$^{-1}$ and $0.69\times10^{-3}$ s$^{-1}$, respectively."

Page 9, line 282-284:
"The CS values were $0.54\times10^{-3}$ s$^{-1}$, $0.74\times10^{-3}$ s$^{-1}$, $0.77\times10^{-3}$ s$^{-1}$, $0.64\times10^{-3}$ s$^{-1}$, and $0.80\times10^{-3}$ s$^{-1}$ for cluster 1, cluster 2, cluster 3, cluster 4, and cluster 5, respectively, indicating that the cluster 1 had the lowest CS."

8. Why GR3-25 while N3-20? Maybe use 3- 25 nm as nucleation mode for consistency with your GR calculations and with previous literature. N3-25 has been referred to as nucleation mode particles in some literature (Vana et al., 2016; Zhou et al., 2020).

Answer: As suggested by the reviewer, $N_{3-25}$, and $N_{25-60}$ were used in texts and related Figures (Figure 2, 4, and 8).

9. Comparison of instruments: how does your nano-smps compare to the instruments at the station? See figure 1 in Kangasluoma et al. (2020).

Answer: We compared our nano-SMPS data with DMPS data at the same station as shown in Figure S1 in the Supplement, suggesting that they were in a good agreement.

Page 6, line 165-167:
"We compared our nano-SMPS data with DMPS data at the same station as shown in Figure S1 in the Supplement, suggesting that they were in a good agreement."

New Figure S1 in the Supplement:

[Figure]

Figure S1. Comparison of monthly average size distributions obtained from the nano-SMPS (3–60 nm) and DMPS (10–810 nm). The error bars indicate standard deviation.

Specific comments:
Line 14: " a higher resolution than ever before", this sentence needs to be changed since previous studies have shown up to 10 s time resolution, unless you mean at the specific location you are measuring. Please change here and elsewhere.

Answer: We originally intended "size" resolution, so we added "size" as follows:
"…a higher size resolution than ever before"

Line 36, anytime, do you mean anytime during the month? or anytime during the day? because very little nighttime NPF that grows to full NPF events are observed in the boundary layer.

Answer: The "anytime" was modified to "anywhere".

Line 38, survivor —> survival
Answer: The "survivor" was changed to "survival".

Line 175, 'Dall Maso' –> Dal Maso.
Answer: We corrected the name.

Line 190, survivable –> survival
Answer: The "survivable" was changed to "survival".

[revised manuscript text omitted]

(a)

[Figure]

(b)

[Figure]

Figure S5. Correlations of daily $N_{3-25}$ versus (a) daily $NH_3$, (b) daily $SO_2$, and (c) daily $H_2SO_4$ concentrations during the measurement period. The dashed line represents a linear regression line with a correlation coefficient (r).

[Figure]

Figure S6. NPF event probability distribution with daily CS and temperature. The cell size was 2 K (temperature) and the ratio of 1.26 between two consecutive CS values.

Table S1. Average concentrations of ionic species ($Na^+$, $Mg^{2+}$, $K^+$, $NH_4^+$, $NO_3^-$, $SO_4^{2-}$, and $Cl^-$) in particulate matter and gaseous species ($NH_3$, $SO_2$, and $H_2SO_4$) in different seasons from 2016 to 2018.

| | Unit | Spring | Summer | Fall | Winter |
|---|---|---|---|---|---|
| $Na^+$ | $\mu g\ m^{-3}$ | 0.27±0.38 | 0.18±0.28 | 0.22±0.28 | 0.31±0.33 |
| $Mg^{2+}$ | $\mu g\ m^{-3}$ | 0.04±0.08 | 0.02±0.04 | 0.03±0.04 | 0.05±0.05 |
| $K^+$ | $\mu g\ m^{-3}$ | 0.05±0.07 | 0.03±0.02 | 0.03±0.02 | 0.03±0.02 |
| $NH_4^+$ | $\mu g\ N\ m^{-3}$ | 0.04±0.05 | 0.02±0.03 | 0.02±0.03 | 0.02±0.02 |
| $NO_3^-$ | $\mu g\ N\ m^{-3}$ | 0.02±0.02 | 0.02±0.02 | 0.02±0.04 | 0.02±0.02 |
| $SO_4^{2-}$ | $\mu g\ S\ m^{-3}$ | 0.19±0.18 | 0.08±0.10 | 0.08±0.09 | 0.11±0.20 |
| $Cl^-$ | $\mu g\ m^{-3}$ | 0.39±0.63 | 0.24±0.43 | 0.35±0.50 | 0.52±0.59 |
| $NH_3$ | $\mu g\ N\ m^{-3}$ | 0.13±0.60 | 0.16±0.22 | 0.10±0.10 | 0.08±0.07 |
| $SO_2$ | $\mu g\ S\ m^{-3}$ | 0.09±0.22 | 0.08±0.11 | 0.08±0.13 | 0.09±0.27 |
| $H_2SO_4$ | $10^5$ molecules $cm^{-3}$ | 7.43±8.16 | 8.59±8.64 | 5.52±8.91 | 0.95±0.69 |

---

## Author Response (AR2)

**Anonymous Referee 3:**

The authors did great improvements to the paper, congratulations. I suggest to the authors to add the measurement period to the methods section, and not only to the results. I also suggest that they replace the sentences 'higher than ever before' appearing in the abstract and main text to the actual size resolution in nm for example 'providing a size distribution of nanoparticles (3–60 nm) with a high size resolution of xx nm'.

Answer: As suggested by the reviewer, we added/clarified the measurement period to the Methods section and related sentences. Also, we omitted 'with a higher size resolution than ever before' in the Abstract, and it was replaced by 'with a lower size limit than before' in the main text, which is what we intended to say.

Page 1, line 13-14 in the Abstract:
"We conducted continuous measurement of nanoparticles down to 3 nm size in the Arctic at Mount Zeppelin, Ny Ålesund, Svalbard, from Oct 2016 to Dec 2018, providing a size distribution of nanoparticles (3–60 nm) ."

Page 3, line 72-75 in the Introduction:
"In this study, we measured number size distribution of nanoparticles down to 3 nm for the first time at Zeppelin station, and obtained continuous size distributions of 3–60 nm particles every 3 min from Oct 2016 to Dec 2018. This allowed the size distribution of nanoparticles to be determined with a lower size limit than before, enabling better identification of whether freshly nucleated particles formed on-site or were transported from other regions after substantial growth."

Page 3, line 89-92 in the Methods:
"The dominant wind patterns (east and southeast from the Kongsvegen glacier (40%) and northwest from the Kongsfjorden channels (14%) during the measurement period (Oct 2016 to Dec 2018)) and elevation suggest that the effects of local sources on the Zeppelin Observatory are small (Beine et al., 2001)."

Page 3, line 357-358 in the Conclusions:
"We examined the characteristics of Arctic NPF at the Mount Zeppelin site by conducting continuous measurements of nanoparticles down to 3 nm size from Oct 2016 to Dec 2018."